# Metabolite exchange between microbiome members produces compounds that influence *Drosophila* behavior

**Caleb N Fischer[1†], Eric P Trautman[2], Jason M Crawford[2], Eric V Stabb[3], Jo Handelsman[1], Nichole A Broderick[1,4,5*]**

[1]Department of Molecular, Cellular and Developmental Biology, Yale University, New Haven, United States; [2]Department of Chemistry, Yale University, New Haven, United States; [3]Department of Microbiology, University of Georgia, Athens, United States; [4]Department of Molecular and Cell Biology, University of Connecticut, Storrs, United States; [5]Institute for Systems Genomics, University of Connecticut, Storrs, United States

**Abstract** Animals host multi-species microbial communities (microbiomes) whose properties may result from inter-species interactions; however, current understanding of host-microbiome interactions derives mostly from studies in which elucidation of microbe-microbe interactions is difficult. In exploring how *Drosophila melanogaster* acquires its microbiome, we found that a microbial community influences *Drosophila* olfactory and egg-laying behaviors differently than individual members. *Drosophila* prefers a *Saccharomyces-Acetobacter* co-culture to the same microorganisms grown individually and then mixed, a response mainly due to the conserved olfactory receptor, *Or42b*. *Acetobacter* metabolism of *Saccharomyces*-derived ethanol was necessary, and acetate and its metabolic derivatives were sufficient, for co-culture preference. Preference correlated with three emergent co-culture properties: ethanol catabolism, a distinct volatile profile, and yeast population decline. Egg-laying preference provided a context-dependent fitness benefit to larvae. We describe a molecular mechanism by which a microbial community affects animal behavior. Our results support a model whereby emergent metabolites signal a beneficial multispecies microbiome.

**\*For correspondence:** nichole. broderick@uconn.edu

**Present address:** [†]Department of Chemistry, Vanderbilt University, Stevenson Center, Nashville, United States

**Competing interests:** The authors declare that no competing interests exist.

## Introduction

Multispecies microbial communities (microbiomes) influence animal biology in diverse ways (*McFall-Ngai et al., 2013*): microbiomes modulate disease (*van Nood et al., 2013*), metabolize nutrients (*Zhu et al., 2011*), synthesize vitamins (*Degnan et al., 2014*), and modify behavior (*Bravo et al., 2011*). A central goal in host-microbiome studies is to understand the molecular mechanisms underpinning these diverse microbiome functions.

Some aspects of microbial community function are the product of inter-species interactions (*Rath and Dorrestein, 2012*; *Manor et al., 2014*; *Gerber, 2014*; *Gonzalez et al., 2012*). For example, microorganisms modulate the metabolomes of neighboring species (*Derewacz et al., 2015*; *Jarosz et al., 2014*) and microbial metabolites (e.g., antibiotics) alter bacterial transcriptional responses (*Goh et al., 2002*). Despite current understanding of microbial inter-species interactions in vitro, some of which has been elucidated in exquisite detail, the consequences of microbial

**eLife digest** Animals associate with communities of microorganisms, also known as their microbiome, that live in or on their bodies. Within these communities, microbes – such as yeast and bacteria – interact by producing chemical compounds called metabolites that can influence the activity of other members of the community. These metabolites can also affect the host, helping with nutrition or causing disease.

The behavior of an animal may help it to acquire its microbiome, although this has not been properly explored experimentally. For example, the fruit fly *Drosophila melanogaster* acquires members of its microbiome from the microbes found on the fermented fruit that it eats. It is possible that the flies – and other animals – respond to microbial metabolites, which act as signals or cues that cause the animal to avoid or seek the microbial community.

The fruit fly microbiome is commonly studied in the laboratory because it has a much simpler composition than mammalian microbiomes. Previous studies have explored how the flies respond to odors produced by individual types of microbes, but none have explored how the behavior of the flies changes in response to the odors produced by a mixed microbial community.

Fischer et al. now show that fruit flies are preferentially attracted to microbiome members that are interacting with each other. The flies detected members of the microbiome by responding to chemicals that are only produced when community members grew together. For example, one member of the microbial community produces ethanol that is then converted to acetate by another community member. Neither ethanol nor acetate alone attracted flies as strongly.

Fischer et al. also discovered that both adult fruit flies and their larvae benefit from acquiring a mixture of different microbes at the same time. Adult flies benefit by avoiding harmful concentrations of either ethanol or acetic acid, and larvae benefit from developing in an environment that reduces how quickly disease-causing microbes can grow.

Overall, the results presented by Fischer et al. detail how flies select a beneficial, interactive microbiome from an external reservoir of microorganisms. Flies also have internal mechanisms, like their immune system, that help them to select their microbiome. Therefore a future challenge will be to integrate the behavioral and internal selection mechanisms into a single model of microbiome acquisition.

interspecies interactions within host-associated microbiomes are just beginning to be explored experimentally.

Insight into host-associated microbiome function has stemmed mostly from whole-microbiome [e.g., re-association of germ-free hosts with whole microbiomes (*Ridaura et al., 2013*) and modeling microbiome function based on gene annotation (*Costello et al., 2009*)] or single-microorganism [e.g., re-association of germ-free hosts with a single microorganism (*Ivanov et al., 2009*)] studies. However, these approaches tend to reveal only limited insight into inter-species microbial interactions, which can provide hosts with essential services. For example, termite symbionts carry genes necessary for metabolism of different parts of complex carbohydrates (*Poulsen et al., 2014*), yet their function has not been demonstrated in vivo; co-occurring human gut symbionts share polysaccharide breakdown products cooperatively (*Rakoff-Nahoum et al., 2014*, *2016*), but the consequences of such interactions for the host are unknown; inter-species bacterial interactions protect *Hydra* from fungal infection (*Fraune et al., 2015*), but the mechanism of host protection is unclear. The need to understand the effects of inter-species microbiome interactions motivated our current work.

Attractive model systems in which to study the outcomes of inter-species microbial interactions for host biology would include a tractable host that harbors a simple multispecies microbiome. Here, we report the use of *Drosophila melanogaster* to study interactions in a simple microbiome and their consequences for host behavior.

The *Drosophila* microbiome consists largely of yeasts, acetic acid bacteria, and lactic acid bacteria (*Chandler et al., 2011*, *2012*; *Broderick and Lemaitre, 2012*; *Camargo and Phaff, 1957*; *Staubach et al., 2013*). *Drosophila* ingests microbiome members from the environment (e.g., fermenting fruit, [*Camargo and Phaff, 1957*; *Barata et al., 2012*; *Erkosar et al., 2013*; *Blum et al.,*

2013; *Broderick et al., 2014*]), a behavior posited as a mechanism for *Drosophila* to select, acquire, and maintain its microbiome (*Broderick and Lemaitre, 2012*; *Blum et al., 2013*). *Drosophila* behavior toward environmental microorganisms has focused on yeasts (*Becher et al., 2012*; *Christiaens et al., 2014*; *Schiabor et al., 2014*; *Palanca et al., 2013*; *Venu et al., 2014*). Yeasts attract *Drosophila* via ester production (*Christiaens et al., 2014*; *Schiabor et al., 2014*), induce *Drosophila* egg-laying behavior (*Becher et al., 2012*), and are vital for larval development (*Becher et al., 2012*). Lactic and acetic acid bacteria produce metabolites (e.g., acids) that may repel *Drosophila* at high acid concentrations, while also inducing egg-laying preference for sites containing acetic acid (*Ai et al., 2010*; *Joseph et al., 2009*). One motivation of our study was to analyze *Drosophila* behavior toward the yeast and bacteria that dominate the *Drosophila* microbiome.

Yeast and bacteria are largely studied within separate *Drosophila* sub-disciplines, despite their shared habitat (*Broderick and Lemaitre, 2012*). Yeasts serve as food, providing *Drosophila* vitamins, sterols, and amino acids (*Broderick and Lemaitre, 2012*). Lactic and acetic acid bacteria are gut microbiome members (*Wong et al., 2011*) promoting larval development (*Shin et al., 2011*; *Storelli et al., 2011*), increasing resistance to pathogens (*Blum et al., 2013*), inducing intestinal stem cell proliferation (*Buchon et al., 2009*), and reducing adult sugar and lipid levels (*Newell and Douglas, 2014*; *Wong et al., 2014*). Since microorganisms that are traditionally considered 'food' co-exist with those considered 'microbiome' in fruit fermentations and the two groups provide *Drosophila* with different resources, we hypothesized that *Drosophila* might detect a beneficial community via metabolites that are produced cooperatively by the desirable symbionts. Alternatively, *Drosophila* might detect a different metabolite as the signal for each symbiont.

Fruit undergoes a well-characterized ripening process in which cell-wall degrading enzymes and amylases convert the firm, starchy tissue into soft, sugar-rich fruit (*El-Zoghbi, 1994*; *Abu-Goukh and Bashir, 2003*; *Mao and Kinsella, 1981*). The high sugar content supports microbial colonization and fermentation by *Drosophila*-associated microorganisms, including yeasts, lactic acid bacteria, and acetic acid bacteria (*Barata et al., 2012*; *Barbe et al., 2001*). *Drosophila* avoids 'green' fruit and is attracted to 'overripe' fruit (*Turner and Ray, 2009*), yet it is unclear how *Drosophila* behavior is influenced by the dynamic multispecies fruit microbiome and its metabolic properties. To this end, we developed a model fruit fermentation system that afforded measurement of microbial populations, microbial metabolites, and *Drosophila* behavior.

Here we demonstrate the importance of emergent microbiome metabolism—quantitatively different or unique metabolites produced by the microbiome, but not by any of its members in isolation—on behavior, suggesting that *Drosophila* larvae and adults benefit by behaviorally selecting a multispecies, interactive microbiome.

## Results

To determine whether *Drosophila* responds to emergent microbial community metabolites, we used the T-maze olfactory assay to analyze *Drosophila* behavioral responses to several *Drosophila* microbiome members grown individually or in communities (*Figure 1A*, *Supplementary file 2*, *Figure 1—figure supplement 1*). When the strains were grown individually, *Drosophila* was strongly attracted to yeasts, moderately attracted to acetic acid bacteria, and neutral or slightly repelled by lactic acid bacteria (*Figure 1B,C*). Because strains within a microbial group attracted *Drosophila* similarly, a representative yeast, acetic acid bacterium, and lactic acid bacterium were used to test the effect of interactions between microbiome members on *Drosophila* behavior. *Drosophila* preferred microbial communities grown together to microorganisms grown individually and then mixed prior to analysis (defined throughout as a separate-culture mixture, *Figure 1D*). Focusing on a model *Saccharomyces cerevisiae* and *Acetobacter malorum* community, we found that when tested against apple juice medium (AJM), *Drosophila* attraction to the community was stronger than to the separate-culture mixture or individual members (*Figure 1E*). In sum, *Drosophila* detects, and prefers, microorganisms growing together to a mixture of the same strains combined after they had completed growth.

We next measured the attractiveness and other properties of the co-culture over time. When grown alone, the microorganisms had similar growth profiles (*Figure 2A*). However, when grown with *A. malorum*, *S. cerevisiae* populations first increased, then decreased between 60 and 72 hr, and were undetectable by 96 hr (*Figure 2A*). The decrease in *S. cerevisiae* viable counts mirrored a decrease in pH (*Figure 2—figure supplement 1A*). *Drosophila* did not prefer the co-culture relative

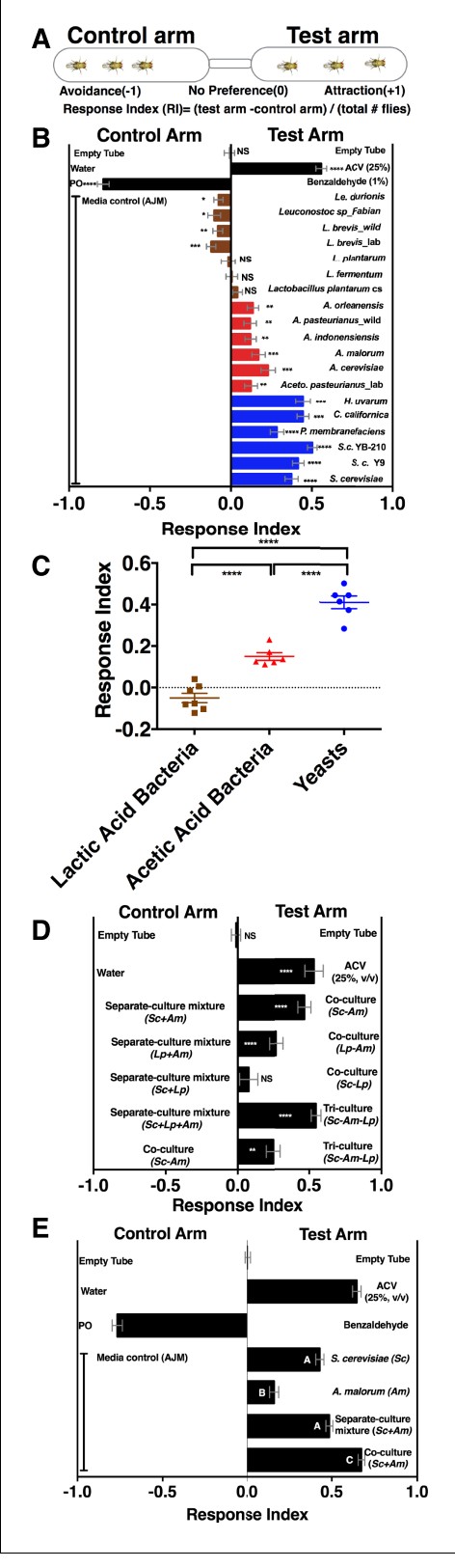

**Figure 1.** *Drosophila* detection of microbe-microbe metabolite exchange. (**A**) T-maze setup and quantification. (**B**) *Drosophila* behavior toward yeasts (blue), acetic acid bacteria (red), and lactic acid

to the separate-culture mixture at 34 hr; however, *Drosophila* was more attracted to the co-culture from 48–127 hr (*Figure 2B*). Moreover, the *Drosophila* attraction to the 96 hr co-culture was stronger than its preference for the 48, 54, or 60 hr co-cultures (*Figure 2B*). *Drosophila* preference for the co-culture correlated with lower pH and *S. cerevisiae* population density, despite *Drosophila* olfactory avoidance of acid (*Ai et al., 2010*) and reliance on yeast for nutrition (*Anagnostou et al., 2010*) (*Figure 2C,D*). *Drosophila* preference did not correlate with viable *A. malorum* populations (*Figure 2—figure supplement 1B*). *Drosophila* preference for the co-culture increased relative to sterile media during 34–96 hr of growth, which is consistent with the increase in *Drosophila* attraction being due to a property of the co-culture rather than to a decrease in attraction to the separate-culture mixture (*Figure 2—figure supplement 1C*). Moreover, *Drosophila* was more attracted to the 72 hr co-culture than individual cultures or the separate-culture mixture at any other growth stage (i.e. 24, 34, 72 hr; *Figure 2—figure supplement 1D,E*). In sum, several properties of the microbial community (e.g. *S. cerevisiae* density, pH) parallel *Drosophila* detection of, and preference for, the co-culture.

Mutants in broadly and narrowly tuned ionotropic and olfactory receptors [Irs and Ors, respectively, (*Abuin et al., 2011*; *Hallem and Carlson, 2006*)] were used to evaluate the role of *Drosophila* olfactory reception in discriminating the co-culture from the separate-culture mixture during and immediately following peak attraction (*Figure 3—figure supplement 1*). During the most attractive phase of the co-culture (e.g. 67–115 hr), homozygous mutants of *Drosophila* ORCO and Or42b showed a significant reduction in attraction to the co-culture, whereas no role was detected for *Drosophila* homozygous mutants in several Irs or Or35a (*Figure 3A*). As co-culture growth proceeded (e.g. 139–163 hr), attraction decreased and the role of Or42b and ORCO waned (*Figure 3*). An independent homozygous mutant of ORCO also showed reduced attraction to the co-culture, whereas the heterozygotes ORCO/+ and Or42/+ were attracted to the co-culture similarly to wild-type flies (*Figure 3B*).

ORCO is a required co-receptor for all other Or gene products (*Larsson et al., 2004*) and Or42b, one of the most conserved olfactory receptors, detects esters and 1,1-diethoxyethane (*Mathew et al., 2013*; *Asahina et al., 2009*; *Stökl et al., 2010*). These results suggest that

*Figure 1 continued*

bacteria (brown) (**Supplementary file 2**). Mean ± SEM of 12–36 replicates (n = 2–6 experiments). Each T-maze replicate uses a technical replicate of a microbial culture and one cohort of *Drosophila* maintained in separate vials for 3–5 days. Mock (two empty tubes), ACV (25% apple cider vinegar versus water), and benzaldehyde (1% versus paraffin oil [PO]). The one-sample t-test was used to assess the mean deviance from 0. Symbols: NS p>0.05; *p≤0.05; **p≤0.01; ***p≤0.001; ****p≤0.0001. (**C**) Mean *Drosophila* behavior toward each microorganism was graphed according to microbial group. The means were compared by one-way ANOVA with Tukey's post-hoc comparison. (**D**) *Drosophila* behavior toward community combinations of a representative yeast, acetic acid bacterium, and lactic acid bacterium in relation to their separate-culture mixture (grown individually and mixed; Sc = S. cerevisiae; Am= A. malorum; Lp = L. plantarum cs) grown for 96 hr; *Drosophila* preference for the three- versus two-membered community is the last column. Mean ± SEM of 12–18 replicates (n = 2–3 experiments). The one-sample t-test assessed the mean deviance from 0. (**E**) *Drosophila* olfactory behavior toward the *S. cerevisiae* and *A. malorum* community and its constituent parts relative to media grown for 48–60 hr. Mean ± SEM of 18–30 replicates (n = 5 experiments). A one-way ANOVA followed by post-hoc Tukey's multiple comparison correction test evaluated whether the means of the experimental groups were different from one another.

The following source data and figure supplements are available for figure 1:

**Source data 1.** Raw *Drosophila* preference data for *Figure 1B,C*.
**Source data 2.** Raw *Drosophila* preference data for *Figure 1D*.
**Source data 3.** Raw *Drosophila* preference data for *Figure 1E*.
**Figure supplement 1.** *Drosophila melanogaster* olfactory behavior toward different culture volumes of *Saccharomyces cerevisiae* and *Acetobacter malorum*.
**Figure supplement 1—source data 1.** Raw Drosophila preference data for *Figure 1—figuresupplement 1*.

Or42b enables *Drosophila* to distinguish the co-culture from the separate-culture mixture. Moreover, a non-ORCO factor explains ~40% of *Drosophila* co-culture preference (**Figure 3A,B**). Previous work found that ORCO is fully responsible for the *Drosophila* attraction to apple cider vinegar (**Semmelhack and Wang, 2009**), suggesting that the behavioral circuit activated by inter-species interactions between *S. cerevisiae* and *A. malorum* is distinct from the circuit activated by apple cider vinegar.

We speculated that the emergent property of co-culture attractiveness might arise from a distinct metabolic profile of the co-culture. Using gas chromatography-mass spectrometry (GC-MS), we identified volatiles unique to or differentially produced in the co-culture compared to the separate-culture mixture. Five co-culture volatiles (ethanol, isobutanol, isoamyl alcohol, acetic acid, isoamyl acetate) were confirmed with standards (**Table 1—source data 1** and **2**) and quantified with standard curves (**Table 1—source data 3** and **4**). The alcohol concentrations were lower, and acetic acid and isoamyl acetate were unique in the co-culture relative to the other experimental groups (**Table 1**). The molecular profile was reminiscent of ethanol catabolism as the unique co-culture metabolic process. We therefore hypothesized that ethanol catabolism was the emergent metabolic process.

We next measured ethanol and acetic acid levels over time (24–156 hr) and compared the chemical dynamics to *Drosophila* preference. Consistent with a relationship between ethanol catabolism, acetic acid anabolism, and *Drosophila* attraction, the dynamics of *Drosophila* co-culture preference mirrored ethanol depression and acetic acid accumulation in the co-culture (**Figure 4A**). Furthermore, as ethanol catabolism and acetic acid anabolism proceeded (36–96 hr), *Drosophila* attraction toward the co-culture increased until 96 hr, at which point it decreased, consistent with lower turnover of ethanol at the end of ethanol catabolism (**Figure 4A**, black line).

We hypothesized that *Drosophila* preferred the community during peak ethanol turnover (e.g. co-cultures at ~72 hr of growth) compared with the community during pre-ethanol catabolism (e.g. co-culture at ~36 hr of growth) or during late-stage ethanol catabolism, in which ethanol turnover is low (e.g. co-culture at ~144

hr of growth; **Figure 4A**). Consistent with our hypothesis, *Drosophila* preferred the co-culture in the middle stage of ethanol catabolism to earlier or later stages of ethanol catabolism (**Figure 4B**). To test directly whether ethanol catabolism underpinned *Drosophila* co-culture preference, we evaluated *Drosophila* preference for the co-culture harboring a mutant in *adhA*, which encodes

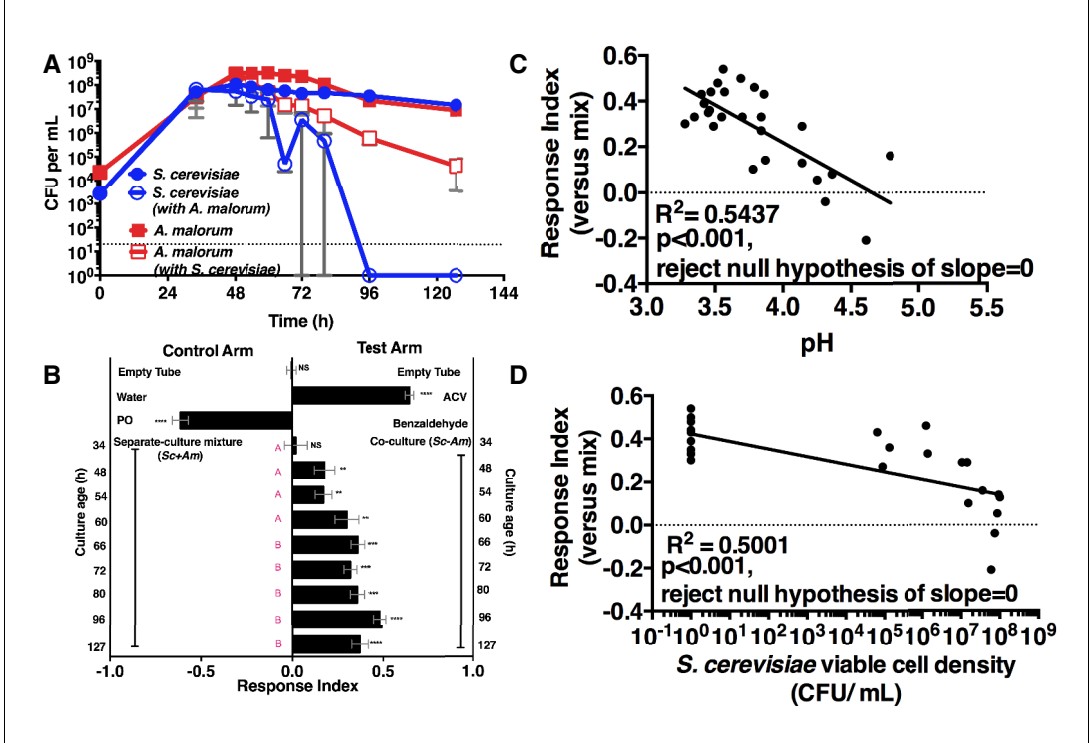

**Figure 2.** *Drosophila* temporal preference for metabolite exchange. (**A**) *S. cerevisiae* and *A. malorum* viable populations. Mean ± SEM of 2–3 experiments with one pooled replicate (2–3 cultures from the same colony) per experiment. Limit of detection is 20 CFU/mL. A curve was fitted to the data with 40 values. Subsequently, an exponential plateau equation was compared between the individual cultures from 0 to 72 hr. The null hypothesis that the k values are the same was not rejected (p>0.05). A separate analysis compared a slope of 0 between *S. cerevisiae* alone and *S. cerevisiae* with *A. malorum* from 48–127 hr. The null hypothesis that the slopes were the same was rejected (p=0.0205). (**B**) *Drosophila* olfactory behavior toward co-cultured *S. cerevisiae* and *A. malorum* versus its separate-culture mixture as a function of culture age. Mean ± SEM of 16–18 replicates from three experiments. Two statistical tests were run. First, a one-sample t-test assessed whether *Drosophila* was attracted, neutral, or repelled by the test arm by evaluating mean deviance from 0. Symbols: NS p>0.05; *p≤0.05; **p≤0.01; ***p≤0.001; ****p≤0.0001. Second, a one-way ANOVA followed by Dunnet's post-hoc multiple comparison test evaluated whether *Drosophila* was attracted to the co-culture aged 96 hr differently than other aged co-cultures. The results are shown in pink; unique letters indicate difference (p<0.05) from 96 hr. (**C**) Relationship between pH and *Drosophila* preference for the *S. cerevisiae* and *A. malorum* co-culture versus the separate-culture mixture. Each data point represents the pH of a co-culture and the mean RI of *Drosophila* toward the same co-culture. A linear standard curve with an unconstrained slope was generated and compared to a null model with slope = 0. The data fit to an unconstrained slope better than to the null model (p<0.0001, slope = −0.3295). (**D**) Relationship between *S. cerevisiae* populations and *Drosophila* preference for the co-culture versus the separate-culture mixture. Each data point represents viable *S. cerevisiae* populations of the culture along with the mean RI value toward the co-culture containing *S. cerevisiae*. A semilog standard curve with an unconstrained slope was generated and compared to a null model with slope = 0. The data fit to an unconstrained slope better than to the null model (p<0.0001, slope = −0.0349).

The following source data and figure supplements are available for figure 2:

**Source data 1.** Raw *Drosophila* preference data for *Figure 2B* & *Figure 2—figure supplement 1C*.

**Source data 2.** Raw *Drosophila* preference data, microbial population data, and pH data for *Figure 2A,C,D* & *Figure 2—figure supplement 1A,B*.

**Figure supplement 1.** Properties of the co-culture and its relationship to *Drosophila* preference.

**Figure supplement 1—source data 1.** Raw *Drosophila* preference data and microbial population data for *Figure 2—figure supplement 1D,E*.

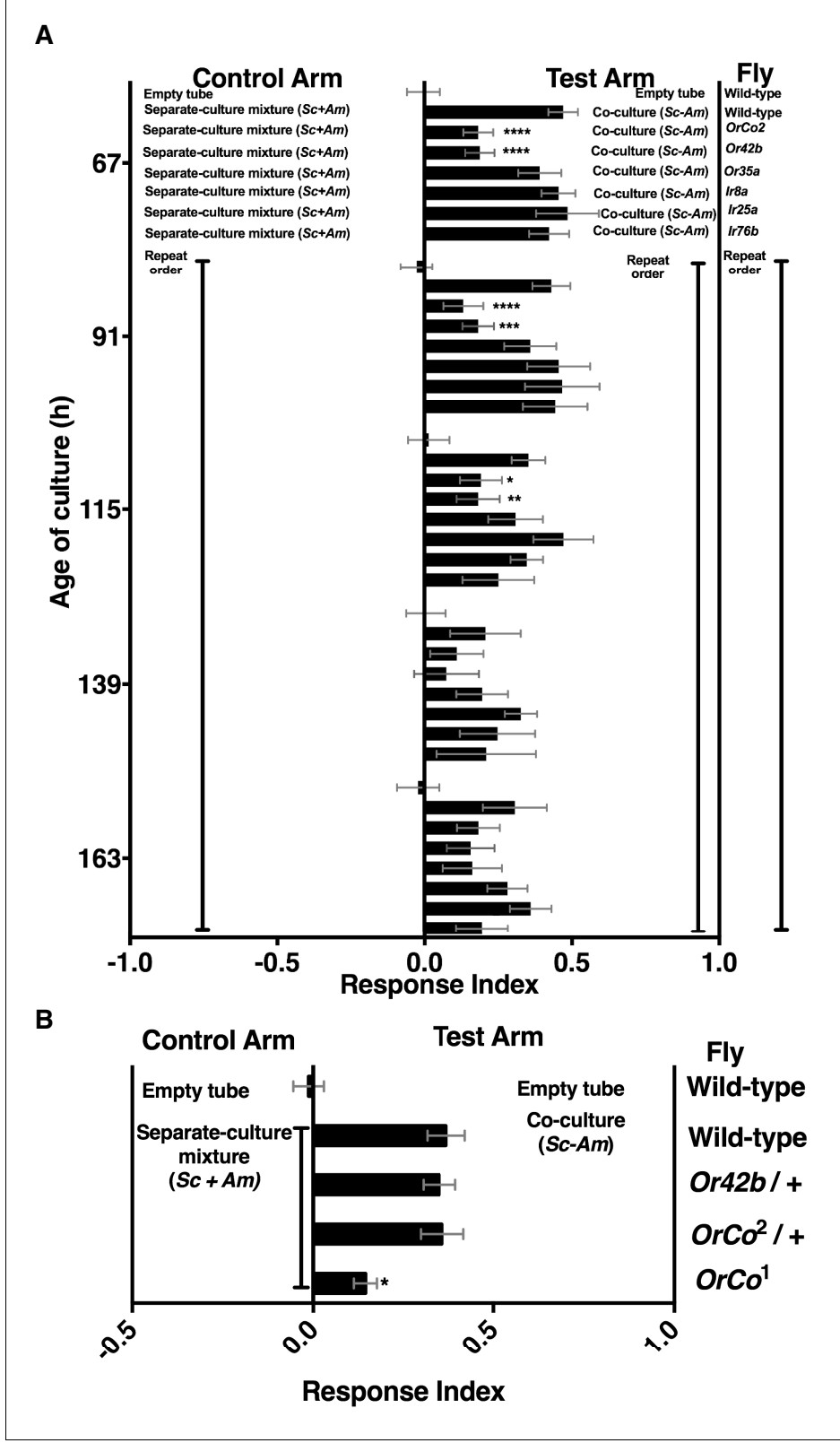

**Figure 3.** Role of olfactory receptor mutants in *Drosophila* detection of inter-species microbial interactions. (**A**) The mean rank of the response index of the various *Drosophila* mutants toward the co-culture was compared with the mean rank of wild-type fly behavior toward the co-culture using the Kruskal-Wallis test followed by Dunn's post-hoc multiple comparisons testing. Symbols: *p≤0.05; **p≤0.01; ***p≤0.001; ****p≤0.0001. A lack of symbol

*Figure 3 continued on next page*

*Figure 3 continued*

indicates no difference when comparing each mutant group to the wild-type group. The behavioral responses of all *Drosophila* (wild-type and each mutant) toward the co-culture was greater than 0 (using the non-parametric Wilcoxon signed rank test in which the medians were compared to 0, p<0.05, no symbols shown). Mean +/- SEM of 12–24 replicates per time point per fly condition (n = 2–4 experiments per time point). (**B**) The mean rank of mutant fly behavior toward the co-culture was compared between wild-type and the specified conditions using the Kruskal-Wallis test followed by Dunn's post-hoc host multiple comparisons testing. Mean +/- SEM of 11–12 replicates (n = 2 experiments).

The following source data and figure supplement are available for figure 3:

**Source data 1.** Raw *Drosophila* preference data and microbial population data for *Figure 3A* and *Figure 3—figure supplement 1*.

**Source data 2.** Raw *Drosophila* preference data for *Figure 3B*.

**Figure supplement 1.** Effect of co-culture age on Drosophila attraction and microbial density.

pyrroloquinoline quinone-dependent alcohol dehydrogenase (PQQ-ADH-I), the enzyme that converts yeast-derived ethanol into acetaldehyde on path to acetic acid (*Shin et al., 2011*). Co-cultures using either *A. malorum* or *A. pomorum* wild-type (WT) along with *S. cerevisiae* were equally attractive to *Drosophila* (*Figure 4—figure supplement 1*). *Drosophila* preferred the co-culture containing *A. pomorum* WT versus a separate-culture mixture; however, *Drosophila* did not prefer the co-culture containing *A. pomorum adhA* versus a separate-culture mixture (*Figure 4C*). Moreover, *Drosophila* preferred the co-culture containing *A. pomorum* WT to the co-culture containing *A. pomorum adhA* (*Figure 4C*). In sum, ethanol catabolism is necessary for *Drosophila* to discriminate between the co-culture and the separate-culture mixture.

We next identified additional metabolites unique to the co-culture using solid-phase microextraction gas chromatography-mass spectrometry (SPME GC-MS). Acetic acid, six acetate esters, an acetaldehyde metabolic derivative (acetoin), a putative acetaldehyde metabolic derivative (2,4,5-trimethyl-1,3-dioxolane), and two unknown metabolites were more abundant in the co-culture relative to the separate-culture mixture or co-culture with *A. pomorum adhA* (*Table 2*, *Table 2—source data 1–6*). To determine the molecular basis for *Drosophila* co-culture preference, select metabolites were added to the co-culture containing *A. pomorum adhA*. Esters and acetic acid, but not esters alone, were sufficient to fully restore the attractiveness of the co-culture containing *A. pomorum adhA* to the co-culture containing *A. pomorum* WT levels (*Figure 4C*).

Although acetate and its metabolic derivatives were sufficient for *Drosophila* co-culture preference, acetaldehyde is a reactive intermediate during ethanol catabolism whose metabolic derivatives might be increased in microbial communities compared with individual microbial cultures. Consistent with this idea, acetoin was moderately increased in the co-culture compared with the separate-culture mixture (*Table 2—source data 1*); strikingly, acetoin was increased ~27 fold in the tri-culture (*S. cerevisiae*, *A. malorum*, and *L. plantarum*) compared to the co-culture (*Figure 5A,B*, *Figure 5—figure supplement 1*) and was attractive to *Drosophila* (*Figure 5C*). In sum, emergent metabolites from two- and three-membered communities, including acetaldehyde metabolic derivatives, attract *Drosophila*.

To further investigate the potential role of acetaldehyde and its metabolic derivatives in *Drosophila* behavior, we performed a dose response in which acetaldehyde was added to the separate-culture mixture (*Figure 6—figure supplement 1A*) to evaluate its ability to induce attractiveness to co-culture levels. Even at the lowest tested levels, acetaldehyde supplementation stimulated the separate-culture mixture to attractiveness levels equal to the co-culture (*Figure 6—figure supplement 1A*). Three acetaldehyde metabolic derivatives—acetoin, 1,1-diethoxyethane (an acetal), and 2,3-butanedione—were sufficient to induce the attractiveness of the separate-culture mixture to levels equivalent to the co-culture containing *A. pomorum* WT using concentrations of each metabolite at or below the physiological concentration of acetoin found in the tri-culture (*Figure 6—figure supplement 1B*).

**Table 1.** Summary of volatiles detected using GC-MS. Relative abundance of volatiles in the co-culture (*S. cerevisiae* and *A. malorum* grown together) compared to the separate-culture mixture (*S. cerevisiae* and *A. malorum* grown separately, and their quantities added in during analysis). GC-MS captured volatiles with XAD-4 beads suspended above the cultures during growth; subsequently, beads were methanol-extracted (n = 6 experiments, **Table 1—source data 1–2**). Quantification is based on two experiments in which a linear regression was computed with standards (**Table 1—source data 3–4**). Quantification is based on beads suspended above the cultures between 84 and 96 hr of culture growth.

| Identity | Standard confirmation | Relative quantification (co-culture: separate-culture mixture) |
|---|---|---|
| Ethanol | Y | 5.0–12.6-fold reduced |
| Isobutanol | Y | 7.3–24.7-fold reduced |
| Isoamyl acetate | Y | unique to co-culture |
| Isoamyl alcohol | Y | 3.6–6.4-fold reduced |
| Acetic acid | Y | unique to co-culture |

Source data 1. Extracted ion chromatograms of five metabolites detected by gas chromatography-mass spectrometry (GC-MS) in **Table 1**. Extracted ion chromatograms of the five metabolites detected by gas chromatography- mass spectrometry (GC-MS). (A) Schematic depicting the experimental setup (B-F) Representative extracted ion chromatograms from one replicate (out of three total) of one experiment (out of 3–4 total) of m/z values corresponding to major metabolites identified in the experimental conditions along with appropriate standards. Acetic acid (B), isoamyl alcohol (C), isoamyl acetate (D) isobutanol (E), and ethanol (F) were identified as the five major metabolites in the co-culture (*S. cerevisiae* and *A. malorum*). Isoamyl alcohol (C), ethanol (E), and isobutanol (F) were identified as the major metabolites in *S. cerevisiae* grown alone. Extracted ion chromatograms were constructed using the m/z value in the title of each graph. For acetic acid and isobutanol, the m/z value used corresponds to the molecular weight of the molecule. For ethanol, the m/z used corresponds to the molecular weight minus one (hydrogen). For isoamyl alcohol, the m/z used corresponds to the loss of the hydroxyl group (depicted), which may have picked up hydrogen and been lost as water. For isoamyl acetate, the m/z value corresponds to the molecule shown within the graph. In all cases, figures showing the complete mass spectra between the metabolite and standard are found in **Source data 2**. Microorganisms were grown 72–96 hr.

Source data 2. Representative spectra of metabolites in **Table 1**. Representative spectra of acetic acid (A-B), isoamyl alcohol (C-E), isoamyl acetate (F-G), ethanol (H-J) and isobutanol (K-L) in standard and experimental samples. Standard concentrations are denoted on individual graphs. All mass spectra are one replicate (out of 3–4 experiments with three replicates per experiment).

Source data 3. Linear regression of metabolites using GC-MS in **Table 1**. Estimation of volatile quantity using GC-MS. Separate experiments are graphed in panels (A-E) and (F-J). (A-E) Data points represent the value of a single replicate per concentration for each standard. The abundance of a single m/z value at a specific retention time was chosen for each standard. The values were fitted with a linear regression and the equation was used to estimate the concentration of the five metabolites in the experimental samples from the same experiment. (F-J) Data points represent the mean ± SEM of three replicates for a given concentration for each standard. The abundance of a single m/z value at a specific retention time was chosen for each standard. The values were fitted with a linear regression. The equation was used to estimate the concentration of the five metabolites in the experimental samples from the same experiment. When applicable an equation was calculated when the line was forced to go through X,Y = 0,0; these equations were used to calculate the concentrations of isoamyl alcohol, isoamyl acetate, and isobutanol.

Source data 4. Raw spectral abundance data as a function of concentration used for linear regressions in **Source data 3**.

A pure metabolite mixture comprised of key metabolic groups produced by microbial communities and identified in this study (esters, acetaldehyde metabolic derivatives, alcohols, acid) attracted *Drosophila* similarly to the co-culture (**Figure 6A,B**). Interestingly, the acetaldehyde metabolic derivatives alone were sufficient to attract *Drosophila* similarly to the co-culture (**Figure 6C**). Moreover, removal of the acetaldehyde metabolic derivatives group alone reduced *Drosophila* attraction (**Figure 6D**). In sum, acetaldehyde metabolic derivatives are potent *Drosophila* attractants.

Overall, our results suggest that both esters and acetaldehyde metabolic derivatives are keystone microbial community metabolites that attract *Drosophila*. We next created a simple 9-metabolite mixture in water (containing only one acid, four esters, and four acetaldehyde metabolic derivatives) and measured *Drosophila* preference toward this mixture in relation to the yeast-acetic acid bacteria co-culture, the yeast-acetic acid bacteria-lactic acid bacteria microbial community, or apple cider vinegar (ACV). The defined mixture used concentrations for each acetaldehyde metabolic derivative similar to the concentration of acetoin in the tri-culture and ester and acid concentrations that were

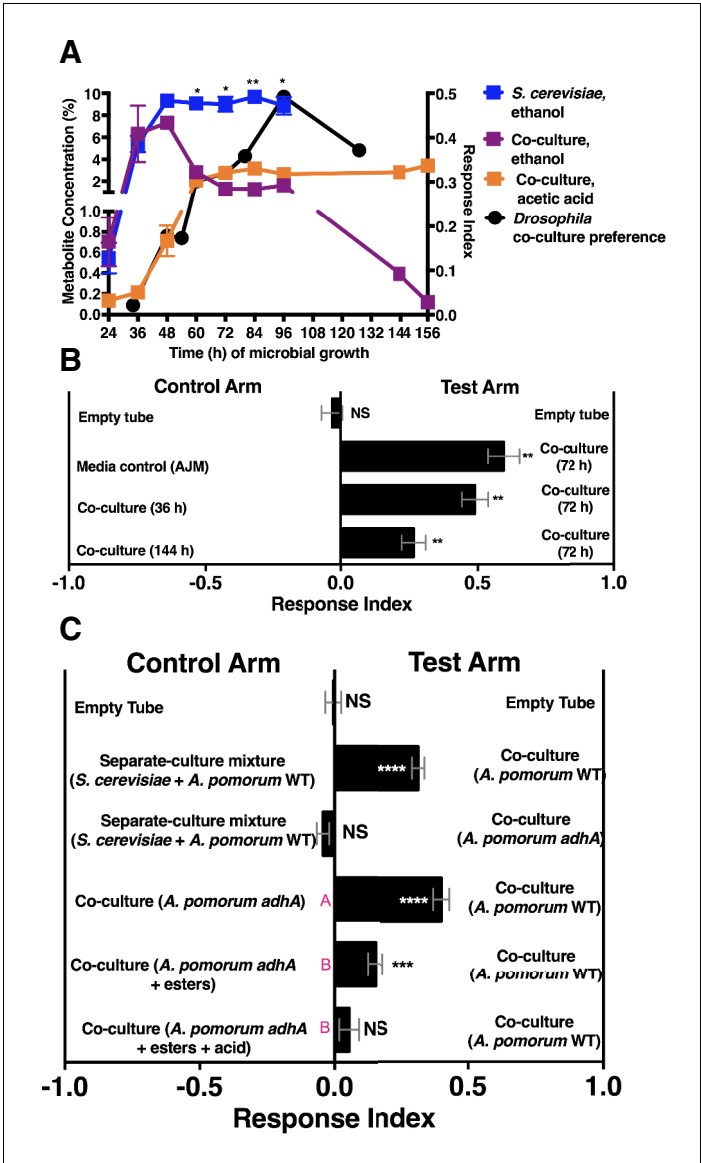

**Figure 4.** *Drosophila* behavior and ethanol catabolism. (A) Dynamics of ethanol, acetic acid, and *Drosophila* co-culture preference. Acetic acid was only detected in the co-culture. The abundance was derived from a linear regression calculated from standards (*Table 1—source data 3*). Chemical data is the mean ± SEM of two values calculated from two experiments with three replicates per experiment (except acetic acid and ethanol concentrations at 144 and 156 hr, which are from one experiment with three replicates). *Drosophila* co-culture preference is the mean value of the preference shown in *Figure 2B*. The estimated ethanol concentrations in the co-culture and *S. cerevisiae* culture were compared with multiple t-tests and multiple comparisons correction by the Holm-Sidak method. Symbols: NS p>0.05; *p≤0.05; **p≤0.01; ***p≤0.001; ****p≤0.0001. (B) *Drosophila* preference for stages of ethanol catabolism. 72 hr is 'mid' stage; 36 hr is 'early' stage and 144 is 'late' stage. The co-culture contains *S. cerevisiae* and *A. malorum* grown for the time indicated. AJM= apple juice media. Data points represent the mean ± SEM of the combined results of two experiments with 8–10 total replicates per group. The one-sample t-test was used to assess the mean deviance from 0. (C) *Drosophila* olfactory behavior toward specified conditions. Mean ± SEM of 2–7 experiments with 10–42 total replicates. Two statistical tests were used to evaluate the behavior. First, a one-sample t-test assessed the mean deviance from 0. Symbols: NS p>0.05; *p≤0.05; **p≤0.01; ***p≤0.001; ****p≤0.0001. Second, a one-way ANOVA with Tukey's post-hoc comparison assessed whether the means of the bottom three experimental groups were different from one another (differences are denoted by unique pink letters). Esters include ethyl acetate, isoamyl acetate, 2-phenethyl acetate, isobutyl acetate, 2-methylbutyl acetate, and methyl acetate; acid is acetic acid. Amounts added are based on

*Figure 4 continued on next page*

*Figure 4 continued*

physiological amounts in co-cultures and are found in *Table 2*. The co-culture contains *S. cerevisiae* and the specified *A. pomorum* strain. acid= acetic acid.

The following source data and figure supplements are available for figure 4:

**Source data 1.** Raw spectral abundance data associated with metabolites graphed in *Figure 4A*.
**Source data 2.** Raw *Drosophila* preference data for *Figure 4B*.
**Source data 3.** Raw *Drosophila* preference data for *Figure 4C*.
**Figure supplement 1.** *Drosophila* behavior toward the co-culture using *A. malorum* or *A. pomorum*.
**Figure supplement 1—source data 1.** Raw *Drosophila* preference data for *Figure 2—figure supplement 1*.

in the range detected in the co-culture. The defined 9-metabolite mixture was more attractive than all other conditions (*Figure 6—figure supplement 2*). In sum, acetaldehyde metabolic derivatives and esters are potent *Drosophila* attractants whose detection may signal the presence of actively metabolizing, multispecies microbial communities.

We hypothesized that *Drosophila* preference for communities during peak ethanol turnover reflected fitness benefits derived from ingesting metabolites associated with different staged communities. To test this hypothesis, we measured adult *Drosophila* survival when given ethanol and acetic acid concentrations characteristic of microbial cultures at different stages of ethanol catabolism. Adult *Drosophila* survival was highest when given metabolites associated with middle-staged ethanol catabolism compared with pre- or end-stage ethanol catabolism (*Figure 7—figure supplement 1A*). In sum, *Drosophila* preference provides benefits associated with consumption of microbial community-generated metabolites.

Next, we explored the relationship between *Drosophila* attraction and egg-laying preference. *Drosophila* preferred to lay eggs in the co-culture containing *A. pomorum* WT to the co-culture containing *A. pomorum adhA* (*Figure 7A*). Therefore, we predicted that *Drosophila* larvae would develop more quickly in the wild-type condition than the *adhA* condition. In contrast, we found that larvae develop more slowly when consuming the co-culture containing *A. pomorum* WT compared with the co-culture containing *A. pomomrum adhA* (*Figure 7B*). This result may be explained by the fact that *A. pomorum* WT kills the nutritious yeast cells, whereas the *A. pomorum adhA* mutant does not (*Figure 7—figure supplement 1B*). Given the role of yeast in *Drosophila* development (*Becher et al., 2012*) the co-culture containing *A. pomorum adhA*, which supports yeast populations, may be more nutritive for developing *Drosophila* larvae than the co-culture containing *A. pomorum* WT.

Another potential selective pressure on the choice of egg-laying sites is the presence of pathogens and parasites. The presence of parasitoid wasps increases *Drosophila* egg deposition in high ethanol concentration sites, which are protective to larvae (*Rollero et al., 2015*). *Drosophila* also avoids laying eggs in habitats containing pathogenic molds by detecting geosmin (*Stensmyr et al., 2012*). Additionally, acetic acid, a unique metabolite in the co-culture containing *A. pomorum* WT, inhibits phytopathogenic fungi (*Kang et al., 2003*). To test whether the co-culture containing *A. pomorum* WT protects developing larvae from environmental fungi, we allowed *Drosophila* to lay eggs in co-culture containing *S. cerevisiae* and either *A. pomorum* WT or *A. pomorum adhA* and quantified the total number of eggs, pupae, and adults. We found that *Drosophila* laid significantly more eggs in the co-culture containing *A. pomorum* WT than the co-culture containing *A. pomorum adhA* (*Figure 7C*). Following open-air exposure to environmental microbes, unidentified fungi grew on the co-cultures containing *A. pomorum adhA*, but did not grow on the co-cultures containing *A. pomorum* WT (*Figure 7D*). Furthermore, more pupae and adults survived in the co-culture containing *A. pomorum* WT compared to the co-culture containing *A. pomorum adhA* (*Figure 7E,F*). In

**Table 2.** Estimated concentrations of key metabolites in the co-culture using SPME GC-MS. Estimated concentrations of differentially concentrated or unique metabolites in the co-culture. Linear regression equations (Lin. reg. eqs. 1 and 2) were estimated from individual experiments in which peak areas of different concentrations of metabolites were fitted with a linear regression (**Table 2—source data 2**, **3**, **5** and **6**). Normalized peak areas correspond to the specified metabolites in co-cultures containing *S. cerevisiae* and *A. malorum*. Separate estimates were derived from a normalized peak area estimated from a single experiment (co-culture and standard samples were from a run with similar internal standard signal) or from the mean normalized peak area estimated from all experiments (co-cultures were run over four days, standards were run on two days). The final estimated concentration was an average of all estimated concentrations (n = 4 estimates (two from each standard regression equation times two estimates of the normalized peak area), except for methyl acetate, n = 2 estimates). The estimated concentrations (except acetoin) were added to the co-culture containing *A. pomorum adhA* (**Figure 4C**). *Ethyl acetate, acetic acid, and acetoin concentrations were estimated from standards (**Table 2—source data 1** and **4**).

| Metabolite | Lin. Reg. eq. 1 | Lin. Reg. eq. 2 | Normalized peak area (single experiment) | Normalized peak area (Average, All experiments) | Estimated concentration (%) |
|---|---|---|---|---|---|
| Isobutyl acetate | Y = 4151X − 0.1319 | Y = 3252X − 0.07251 | 0.29 | 1.16 | 0.00023 |
| Isoamyl acetate | y = 8158X | Y = 7800X | 0.78 | 3.8 | 0.00026 |
| 2-Phenethyl acetate | Y = 5129X −0.04011 | Y = 6972X −0.2013 | 1.2 | 1.9 | 0.00028 |
| 2-Methylbutyl acetate acetate | Y = 8995X − 0.05042 | Y = 8087X−0.1307 | 0.56 | 3.1 | 0.00023 |
| Methyl acetate | Y = 75.22X +0.004457 | NA | 0.018 | 0.040 | 0.00033 |
| Ethyl acetate | NA | NA | NA | NA | ~0.02* |
| Acetic acid | NA | NA | NA | NA | ~3.0* |
| Acetoin | NA | NA | NA | NA | ~0.01* |

Source data 1. Extracted ion chromatograms of differentially emitted or unique metabolites in the co-culture in *Table 2*. Extracted ion chromatograms of differentially emitted or unique metabolites in the co-culture according to solid phase microextraction gas chromatography-mass spectrometry (SPME GC-MS). Specific metabolites are displayed above each panel. For each panel, the left-most plot compares the co-culture containing *S. cerevisiae* and *A. malorum* to *S. cerevisiae* grown alone, *A. malorum* grown alone, or media (AJM [apple juice medium]); the right-most plot compares the co-culture containing *S. cerevisiae* and *A. pomorum* wild-type to the co-culture containing *S. cerevisiae* and *A. pomorum adhA*, since *A. pomorum adhA* is required for *Drosophila* co-culture preference (**Figure 5A**). The two plots within the same panel contain the same standard. The y-axis for each plot is the ion current for a m/z value that discriminates the metabolite of interest over a specific retention time window. The following m/z values were chosen for each metabolite based on standards or, in the cases of putative and unknown metabolites (I and J) were chosen from the experimental groups: (A) m/z 74.04 (B) m/z 88.08 (C) m/z 73.03 (D) 87.05 (E) 74.02 (F) 104.04 (G) 60.05 (H) 88.05 (I) 101.06 (J) 101.06. Each panel is one representative replicate of 1 experiment (out of 3–5 total replicates in three experiments).

Source data 2. Linear regression of metabolites in defined metabolite mixtures in *Table 2*. Normalized peak areas corresponding to metabolites in a defined metabolite mixture (from SPME GC-MS). A linear regression was calculated to quantify the metabolites in the co-culture. Each concentration is from one replicate. A-E and F-I are two separate experiments. Linear regression was used to estimate the concentration of the metabolites in the co-culture containing *S. cerevisiae* and *A. malorum* (**Table 2**) and to complement the co-culture containing *A. pomorum adhA* (**Figure 4C**).

Source data 3. Peak area as a function of concentration used to estimate metabolite concentrations in co-cultures in *Table 2*.

Source data 4. Extracted ion chromatograms of various m/z values used in.

Source data 5. Peak areas as a function of metabolite concentration used in linear regression in *Source data 2A–E*.

Source data 6. Peak areas as a function of metabolite concentration used in *Source data 2F–I*.

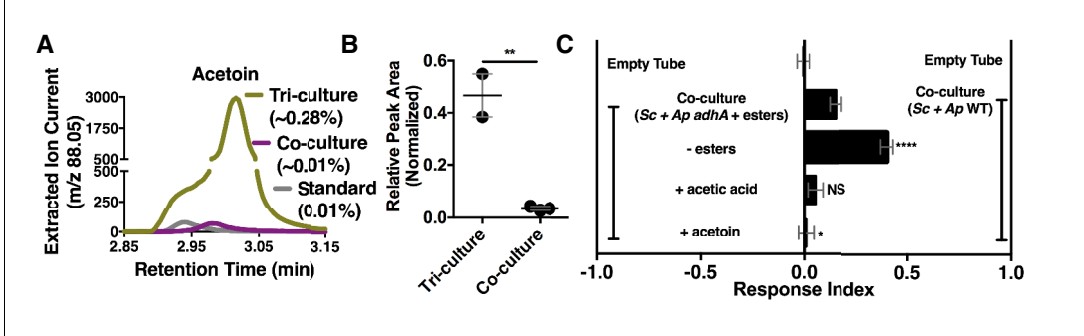

**Figure 5.** Acetaldehyde metabolic derivatives as attractive microbial community generated metabolites. (**A**) Representative chromatogram of m/z 88.05 in the tri-culture (*S. cerevisiae-A. malorum-L. plantarum*) compared to the co-culture (*S. cerevisiae* and *A. malorum*). (**B**) Estimated quantification is based on a linear regression of acetoin (*Figure 6—figure supplement 1*). Relative quantification of acetoin in the tri-culture (one replicate with *A. malorum* and one replicate with *A. pomorum* from separate days) and the co-culture (one replicate with *A. malorum* and two replicates with *A. pomorum* from separate days). Difference in peak areas was assessed by an unpaired two-tailed t-test (\*\*$p \leq 0.01$). (**C**) Mean ± SEM of three experiments with 16–18 total replicates. A one-way ANOVA with Tukey's post-hoc multiple comparisons correction assessed the differences between *Drosophila* behavior toward the co-culture with *A. pomorum adhA* and esters to various groups in which individual molecular groups were removed or added ($p > 0.05$; \*$p \leq 0.05$; \*\*$p \leq 0.01$; \*\*\*$p \leq 0.001$; \*\*\*\*$p \leq 0.0001$). Esters include ethyl acetate, isoamyl acetate, 2-phenethyl acetate, isobutyl acetate, 2-methylbutyl acetate, and methyl acetate. Esters added are based on physiological amounts in co-cultures and are calculated in *Table 2* and *Table 2—source data 2*). Acetoin is added in a similar amount as the tri-culture. *Sc = S. cerevisiae*, *Ap = A. pomorum*.

The following source data and figure supplements are available for figure 5:

**Source data 1.** Extracted ion current for m/z 88.05 in *Figure 5A*.

**Source data 2.** Peak areas associated with acetoin for *Figure 5B*.

**Source data 3.** Raw *Drosophila* preference data for *Figure 5C*.

**Figure supplement 1.** Acetoin linear regression.

**Figure supplement 1—source data 1.** Extracted ion current for *Figure 5—figure supplement 1*.

sum, *Drosophila* egg-laying preference in the co-culture containing *A. pomorum* WT may reflect an underlying benefit in fungal pathogen defense.

## Discussion

Here, we have demonstrated how emergent properties of a microbial community—volatile profile, population dynamics, and pH—influence *Drosophila* attraction, survival, and egg-laying behaviors. Our study is the first to identify the consequences of microbe-microbe metabolic exchange on animal behavior and discovers additional microbial interactions that attract *Drosophila* for further mechanistic study (*Figure 1D*).

Microbe-microbe metabolic exchange generates unique and quantitatively different volatiles from those resulting from individual microbial metabolism (*Table 1 and 2*, *Figure 8*). *Acetobacter*-generated acetate coupled to *Saccharomyces*-derived alcohols spawn diverse acetate esters (*Table 1 and 2*). We hypothesize that more complex and diverse communities, comprising alcohol-producing yeasts, acetate-producing *Acetobacter*, and lactate-producing *Lactobacillus*, will generate a wider array of attractive esters (*Figure 8*). The community of *S. cerevisiae*, *A. malorum*, and *L. plantarum* emitted higher levels of acetoin and attracted *Drosophila* more strongly than the co-culture of *S. cerevisiae* and *A. malorum* (*Figure 1D*, *Figure 5*). Acetoin and 2,3-butanedione are formed by an α-acetolactate intermediate in bacteria and directly from acetaldehyde in yeast (*Chuang et al., 1968*). We

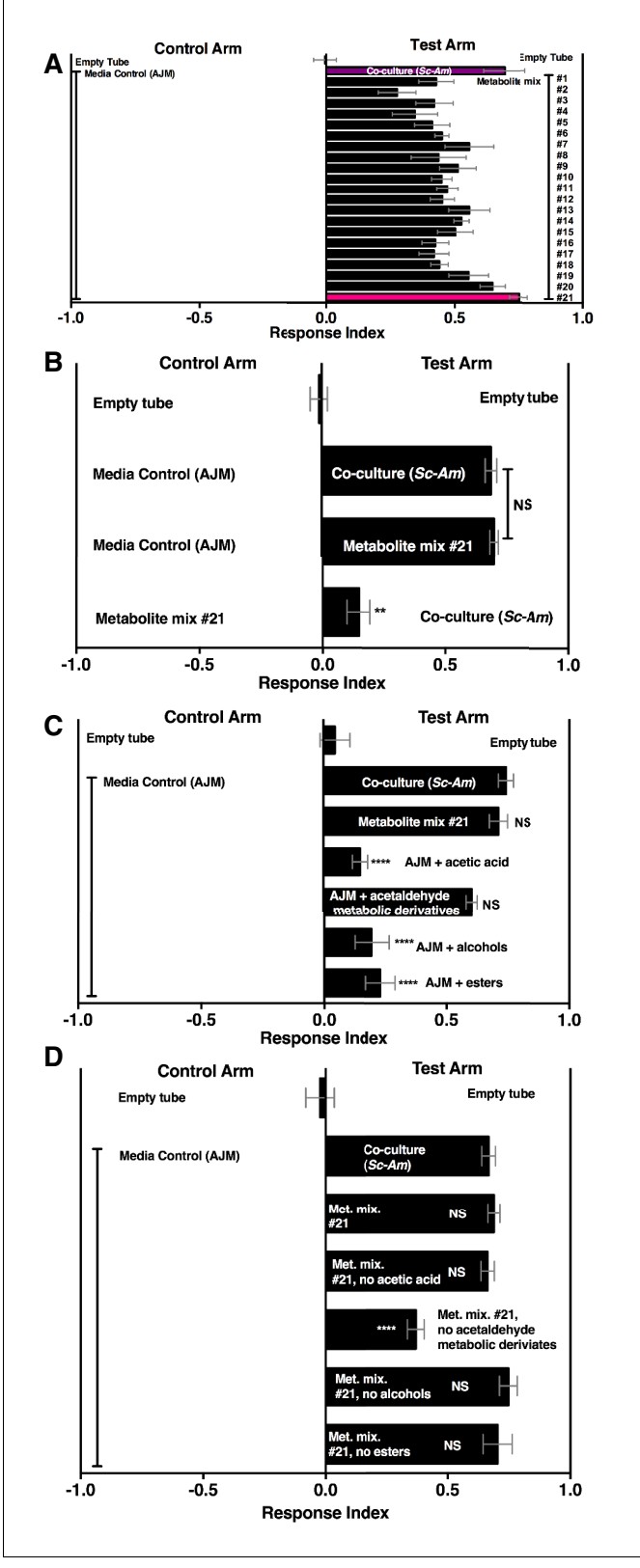

**Figure 6.** *Drosophila* behavior toward 21 metabolite mixtures . (**A**) *Supplementary file 5* contains the concentrations of all mixtures (in 50% AJM). The co-culture was grown for 96 hr. Mean ± SEM of 4–6 replicates per experimental group. Groups were tested over five days. (**B**) *Drosophila* attraction to a co-culture grown for 96 hr and metabolite mixture #21. Mean ± SEM of three experiments with 17–18 replicates per group. A Mann-Whitney

*Figure 6 continued on next page*

*Figure 6 continued*

test compared the median values of the co-culture and metabolite mixture #21; the Wilcoxon Signed-Rank test compared the median value of fly behavior toward the co-culture relative to metabolite mixture #21 to 0. (**C**) Sufficiency of metabolite groups to attract *Drosophila*. The individual groups are: acetaldaldehyde metabolic derivatives (1,1-diethoxyethane; acetoin; 2,3-butanedione); alcohols (ethanol; isobutanol; isoamyl alcohol; 2-methyl, 1-butanol; benzeneethanol); esters (isoamyl acetate; ethyl acetate; isobutyl acetate; 2-phenethyl acetate; butyl acetate; 2-methylbutyl acetate; methyl acetate; phenethyl benzoate; propyl acetate; ethyl isobutyrate; ethyl hexanoate; isovaleric acid; butyl ester; ethyl octanoate; ethyl decanoate; ethyl laurate); and acetic acid (acetic acid). Mean ± SEM of 6 replicates of 1 experiment (except the acetaldehyde metabolic derivative group which is 12 replicates from two experiments). A one-way ANOVA followed by Dunnet's post-hoc comparison assessed the difference between the co-culture and all experimental groups. NS p>0.05; *p≤0.05; **p≤0.01; ***p≤0.001; ****p≤0.0001 (**D**) The same groups used in **C** were used and removed from metabolite mix #21. The difference between the co-culture (*Sc-Am*) and each group was assessed in the same manner as in **C**. Mean +/- SEM of 6 replicates from one experiment.

The following source data and figure supplements are available for figure 6:

**Source data 1.** Concentrations of mixtures and raw *Drosophila* preference data for *Figure 6*.

**Figure supplement 1.** Acetaldehyde metabolic derivatives can complement the co-culture containing *A. pomorum adhA*, although their physiological concentrations are unknown.

**Figure supplement 1—source data 1.** Raw *Drosophila* preference data for *Figure 6—figure supplement 1*.

**Figure supplement 2.** *Drosophila* behavior toward water amended with nine metabolites (9-metabolite mixture) versus three different apple cider vinegars (ACV), a co-culture (*Sc-Am* = *S. cerevisiae* and *A. malorum*), or tri-culture (*Sc-Am-Lp* = *S. cerevisiae*, *A. malorum*, *L. plantarum* cs).

**Figure supplement 2—source data 1.** Raw *Drosophila* preference data for *Figure 6—figure supplement 2*.

therefore hypothesize that communities of yeasts and bacteria may emit high levels of attractive acetaldehyde metabolic derivatives (*Figure 8*).

Previous studies have found that yeasts alone can produce esters in high concentrations (*Becher et al., 2012*; *Christiaens et al., 2014*; *Schiabor et al., 2014*). In this study, we found that *S. cerevisiae* produced low quantities of esters when grown alone. One explanation for the low ester production is that in contrast to previous studies that have used more complex media, we used an apple juice medium that is much lower in nitrogen content. Nitrogen content positively correlates with the yeast ester production (*Becher et al., 2012*; *Rollero et al., 2015*). Our results suggest that environmental nitrogen availability might predict microbial ester production and *Drosophila* attraction. In high nitrogen environments, yeasts likely produce ester compounds and strongly attract *Drosophila*. However, in low nitrogen environments *Acetobacter* may be responsible for ester production and *Drosophila* attraction; *Acetobacter* may be capable of producing esters in low nitrogen conditions or may generate locally high nitrogen environments by assimilating nitrogen from yeast killed by its production of acetic acid. Future work should determine the relationship in wild fruit fermentations between nitrogen content and ester production by yeasts and bacteria.

*Drosophila* behavioral studies have mostly focused on yeasts. Yeasts attract *Drosophila* and are the preferred substrate for *Drosophila* to lay eggs (*Becher et al., 2012*). However, we find that *Drosophila* attraction toward the co-culture increases as yeast viability declines (*Figure 2*). One reason why *Drosophila* might be attracted to the co-culture as yeast populations decline is that yeasts provide essential nutrients. As such, the lysis of viable yeast by *Acetobacter* may benefit *Drosophila* through the liberation of nutrients. An alternative explanation is that in their interaction with *Drosophila*, *Acetobacter* may have benefited by evolving to produce esters that in other contexts (e.g. high nitrogen environments) are produced by yeasts. The contribution of *Drosophila*-associated bacteria to *Drosophila* behavior is not as well understood as yeasts (*Venu et al., 2014*). Our results

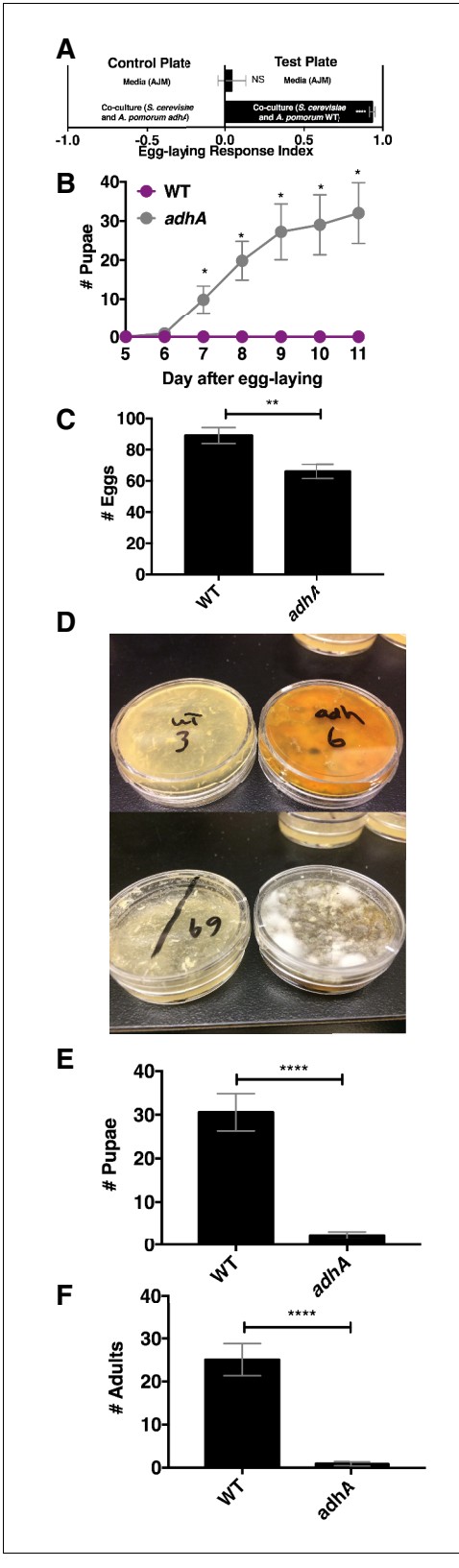

**Figure 7.** *Drosophila* egg-laying preference, nutrition, and pathogen protection. (**A**) *Drosophila* was given a choice to lay eggs in a co-culture containing *S. cerevisiae* and *A. pomorum* wild-type (WT) or *S.*

suggest that non-yeast microorganisms, especially when grown in microbial communities, affect *Drosophila* behaviors. We reason that additional studies that couple chemical microbial ecology with *Drosophila* behavior will herald the discovery of additional microbe-influenced behaviors and microbial community-generated metabolites.

This study demonstrates the coordination of ethanol synthesis and catabolism by *S. cerevisiae* and *Acetobacter*, respectively, and the role of ethanol in *Drosophila* behavior and survival. Non-*Saccharomyces Drosophila* microbiome members also make ethanol (*Ruyters et al., 2015*) and diverse acetic acid bacteria catabolize ethanol, generalizing our findings to other microbial community combinations. Ethanol can have deleterious or beneficial fitness consequences for *Drosophila* depending on concentration (*Ranganathan et al., 1987*; *Azanchi et al., 2013*) and ecological context (*Kacsoh et al., 2013*). Our results are consistent with *Drosophila* using products of inter-species microbiome metabolism to detect a community that titrates ethanol concentration optimally for the host. Work that further dissects the consequences of acetic acid and ethanol concentrations on *Drosophila* biology and investigates other community-level metabolic profiles will be of interest to enrich the chemical and ecological portrait of the *Drosophila* microbiome.

*Drosophila* egg-laying preference for the co-culture containing *A. pomorum* WT may provide a fitness tradeoff for the host. On the one hand, we observed that juice agar plates inoculated with the co-culture containing *A. pomorum* WT had fewer viable yeast cells and larvae developed more slowly, likely due to the lower vital nutrients (e.g. protein, vitamins) than would be available in the co-culture containing *A. pomorum adhA*. On the other hand, when exposed to environmental microbes, juice agar plates inoculated with the co-culture containing *A. pomorum* WT were not invaded by fungi, whereas the co-culture containing *A. pomorum adhA* was susceptible to fungal growth. This suggests that in more natural conditions the catabolism of ethanol into acetic acid, which delays larval development in the microbial community studied here (e.g. in a community with *S. cerevisiae*), ultimately has a protective effect. Whether this is due to a direct elimination of pathogens or instead indirectly limits fungal competition, as has been shown for dietary yeasts and *Aspergillus* sp. (*Rohlfs and Kürschner, 2010*) is unknown. Future work that more thoroughly dissects the *Drosophila* fitness

*Figure 7 continued*

*cerevisiae* and *A. pomorum adhA*. The co-cultures were grown for 96 hr and mixed 1:1 with a 1.6% agarose solution. *Drosophila* was allowed to lay eggs for eight hours. The Wilcoxon signed-rank test evaluated whether the median value of each experimental group was different from 0. Mean ± SEM of 16–18 replicates from two experiments. (**B**) *Drosophila* (40 females and 15 males) deposited eggs in fly vials for 4 hr containing the co-culture of *S. cerevisiae* and *A. pomorum* WT (WT) or the co-culture of *S. cerevisiae* and *A. pomorum adhA* (adhA). Subsequently the number of pupae in each condition was monitored over time. Mean ± SEM of 5 replicates of 1 experiment. Between 12–16 d, larvae pupated in 3/5 WT replicates. Multiple unpaired t-tests were used to compare means at each time point. *p<0.05. (**C**) *Drosophila* (40 females and 15 males) deposited eggs for 4 hr after which the total number of eggs were counted in the two experimental groups. Mean +/- SEM of 12 replicates of 1 of 2 representative experiments. A Mann-Whitney test compared the medians of each group. NS p>0.05; *p≤0.05; **p≤0.01; ***p≤0.001; ****p≤0.0001. (**D**) three days after egg-laying the plates containing eggs quantified in (**C**) were exposed to the open environment and the consequence of exposure was the growth of unidentified fungi, as pictured. Control plates that were not exposed to the environment did not harbor any fungi. In experiment 1, 12/12 of the *adhA* plates harbored fungi and 0/12 plates of WT plates harbored fungi. In the second experiment 4/6 *adhA* plates harbored fungi and 0/6 of WT plates harbored fungi. (**E, F**) Following environmental exposure, the eggs were followed through pupation (**E**) and adulthood (**F**). Mean +/- SEM of 12 replicates of 1 of 2 representative experiments. The median values in E and F were compared the same way as in **C**.

The following source data and figure supplements are available for figure 7:

**Source data 1.** Raw *Drosophila* egg-laying preference data for *Figure 7A*.

**Source data 2.** Raw developmental data for *Figure 7B, C,E,F*.

**Figure supplement 1.** Impact of co-culture metabolites on adult survival and yeast populations.

**Figure supplement 1—source data 1.** Raw survival proportions for *Figure 7-figuresupplement1A*.

tradeoffs that result from its association with different microbiomes is of interest.

Our work raises questions about the consequences of the observed behavior on microbiome assembly and stability in the *Drosophila* intestine. *Drosophila* possesses specific and regionalized gut immune responses to the microbiome (*Lhocine et al., 2008*; *Ryu et al., 2008*; *Paredes et al., 2011*; *Costechareyre et al., 2016*) implying a tolerant environment in which privileged microbiome members are maintained and reproduce in the *Drosophila* intestine. Other work suggests that *Drosophila* acquires its adult microbiome from exogenous sources, that adult microbiome abundance drops without continuous ingestion of exogenous microorganisms, and that the microbiome can be shaped by diet (*Chandler et al., 2011*; *Blum et al., 2013*; *Broderick et al., 2014*). As such, a combination of internal mechanisms, exogenous factors, and host behavior likely sculpt the microbiome; determining the relative contribution of each will be important moving forward. Complicating our understanding of the contribution of these factors is the opaque distinction between 'microbiome' and 'food', since both are ingested from the environment (*Broderick, 2016*). To dissect the formation and stability of the *Drosophila* microbiome, the fate of ingested microorganisms needs to be monitored and microbial intestinal replication needs to be surveyed as a function of *Drosophila* behavior, age, immune status, microbiome membership, and nutritional state [e.g. using synthetic diets without yeast; (*Shin et al., 2011*; *Piper et al., 2014*)].

In sum, our results support a model in which the *Drosophila* olfactory system is tuned to fruity (e.g., esters) and buttery (several acetaldehyde metabolic derivatives, such as 2,3-butanedione) smelling metabolites promoted by microbe-microbe interactions. We anticipate that accounting for microbial interactions in diverse host-microbe studies will lead to new insights into diverse aspects of microbial-animal symbioses.

## Materials and methods

### Fly maintenance

Fly stocks, genotypes, and sources are listed in *Supplementary file 1*. *Drosophila melanogaster* was reared at 25°C on a 12 hr:12 hr light: dark cycle on autoclaved food (5% yeast, 10% dextrose, 7% cornmeal, 0.6% propionic acid, 0.7% agar).

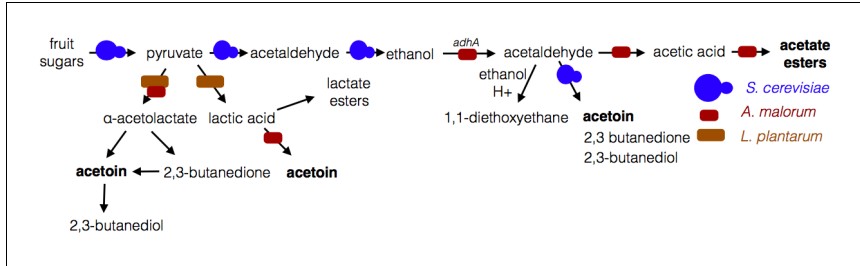

**Figure 8.** Model of microbe-microbe metabolite exchange. Bolded are metabolites increased due to microbe-microbe interactions.

## Microbial strains

Microorganisms used in this study are listed and described in *Supplementary file 2*. Microorganisms were streaked onto yeast-peptone dextrose (YPD; 1% yeast extract (Becton Dickinson, and Company, Franklin Lakes, NJ, USA), 2% peptone (Becton Dickinson, and Company, Franklin Lakes, NJ, USA), and 2% dextrose [Avantor Performance Materials, Center Valley, PA, USA]) or Man, de Rosa, Sharpe (MRS, Fisher Scientific, Waltham, MA, USA) plates from a freezer stock.

## T-maze olfactory attraction assays

The T-maze apparatus was a kind gift of the Carlson Laboratory. Flies were wet-starved for 15–26 hr prior to T-maze olfactory experiments by placing flies into vials containing Kimwipes (Kimberly Clark, Dallas, TX, USA) soaked with 2 mL of milliQ water. Flies were collected within four days (<65 flies per vial) of emergence and matured on autoclaved food. Flies between 3 and 10 days-old were used in experiments.

Single microbial colonies were picked from rich media (MRS and YPD) plates and grown overnight. Cultures were washed 1X in PBS, diluted 100-fold, and 10 µl was aliquoted into 3 mL of apple juice media (AJM, apple juice (Martinelli's Gold Medal, Watsonville, CA, USA), pH adjusted to 5.3 with 5M NaOH, with 0.5% yeast extract). Media was filtered with a 0.22 µM-size pore attached to a 250 mL polystyrene bottle (Corning, NY, USA). For co-culture experiments, 1e3-1e5 CFU of each microorganism was placed simultaneously into AJM. Microorganisms were grown in 14 mL round bottom polypropylene tubes (Corning Science, Tamaulipas, Mexico) at 28°C, 200 rpm for the time noted in individual experiments. The microbial culture was diluted 1:1 with sterile milliQ water (0.22 µM filter [Millipore, Billerica, MA, USA]) and placed directly onto autoclaved 10 mm round Whatman filter paper (GE Healthcare Life Science, Pittsburgh, PA, USA) placed near the bottom of 15 mL CentriStar centrifuge tubes (Corning, NY). A total volume of 10 µl was used for all experiments.

Tubes containing 10 µl of total volume (1:1 microbial culture: water) placed onto 10 mM filter paper and *Drosophila* were placed into the behavioral room (20–25°C, 50–70% humidity maintained by a humidifier (Sunbeam Tower Humidifier, Boca Raton, FL, USA) and equilibrated for 10 min prior to the beginning of the experiment. Flies (~40–130) were knocked into the T-maze apparatus and rested for ~1 min. Subsequently, the two arms of the T-maze were twisted into the T-maze apparatus and the flies were allowed to choose from the test and control arms for 2 min in the dark. No airflow was used in the T-maze assay. Troubleshooting experiments in which red light was used to observe *Drosophila* behavior suggested that *Drosophila* stopped short of reaching the culture placed on the filter paper at the end of the tube. The test arm was alternated from one side of the apparatus to the other every experimental replicate. A Response Index (RI) was computed to analyze preference for the test arm (flies in test arm-flies in control arm)/(total flies).

## Chemicals

Chemicals can be found in *Supplementary file 3*.

## Microbial populations and pH

Selective plates were used to distinguish *S. cerevisiae* from *A. malorum*. MRS containing 50 µg/mL cycloheximide selected for *A. malorum* while MRS containing 10 µg/mL chloramphenicol and 20 µg/mL tetracycline selected for *S. cerevisiae*. pH of filtered cultures (0.22 µM) was measured using a Beckman Coulter pH meter (Model Phi510, Fullerton, CA, USA).

## Gas chromatography- Mass spectrometry

Microbial samples were grown in AJM for a specified amount of time in 14 mL round bottom tubes fitted with an autoclaved tissue strainer (250 µM nylon mesh (Thermo Scientific Pierce, Grand Island, NY) holding between 0.03 and 0.05 grams of autoclaved Amberlite XAD-4 resin (Sigma-Aldrich, St Louis, MO, USA) prewashed in water and methanol. After microbial growth, XAD-4 from two cultures was dumped into an autoclaved glass vial. XAD-4 was swirled with 900 µl methanol for 30 s. 500–750 µl of methanol was removed for GC-MS analysis. Quantification for *Table 1* was derived from beads suspended above the cultures from 84–96 hr of growth. Quantification for *Figure 4A* was derived from beads suspended above the culture every 12 hr; time points on the graph refer to the end point of the 12 hr span (e.g. 84 hr corresponds to beads suspended from 72–84 hr of growth).

Samples (5 µl of methanol-extracted samples) were injected into the GC-MS (Agilent 7890A/5975C) at 250°C using helium as the carrier gas at a flow rate of 1.1 mL per minute (column head pressure 13 psi). The following chromatography temperature program was used for experiments to initially identify metabolites in the co-culture and individually grown microorganisms: 40°C for 3 min ramped at 1.7°C per minute to 200°C (held for 3 min) then to 220°C at 3°C per min and held for a further 5 min. The total run time was 111.78 min. For experiments focused on the five major metabolites, a shorter program was used that maintained the same first 10 min of the previous method (all five volatiles eluted within 9 min). The chromatography temperature program was 40°C for 3 min, ramped at 1.7°C per min to 46.8°C and held for 3 min, then ramped at 60°C per min until 220°C and held for 5 min. The total run time was 17.9 min.

The mass spectrometer was run in electron-impact (EI) mode at 70 eV. The temperatures of the transfer line, quadrupole, and ionization source were 220°C, 180°C, and 230°C respectively. The ionization was off during the first 4 min to avoid solvent overloading with a source temperature of 230°C. Mass spectra were recorded in the range of 35–300 m/z. Full scan mode was used at a scan rate of 6 scans/sec. The electron multiplier voltage was set in the relative mode to autotune procedure.

In the initial experiments peaks were manually picked using Agilent Chemstation Software. Volatiles associated with peaks were searched against the National Institute of Standards (NIST) 11 database. Subsequent experiments focused on the five major volatiles identified in the initial experiments by performing extracted ion chromatograms using an ion that successfully identified a standard at a specific retention time. Quantification was performed by tabulating the maximum abundance of the ion at a characteristic retention time and using a linear regression equation from a dose-response of the standards (*Table 1—source data 3* and *4*).

## Headspace solid phase microextraction (SPME) Gas chromatography-Mass spectrometry

A Waters GCT Premier gas chromatography time of flight mass spectrometer (Milford, MA) with a DB-5MS column (30m x 0.25 mm ID x 0.25 µm film thickness; Agilent) was used. Live cultures were transferred to autoclaved glass vials (20 mL, 23×75 mm, Supelco, Bellefonte, PA, USA) with screw caps (18 mm, 35 Shore A, Supelco, Bellefonte, PA, USA) after growing for 72 hr.

The glass vials containing live microbial cultures were analyzed via a 50/30 µm carboxen/divinylbenzene/polydimethylsiloxane Stableflex solid-phase micro-extraction (SPME) fiber. The extraction methodology was based on previous studies using SPME to extract volatiles form vinegars (*Callejón et al., 2008*; *Xiao et al., 2011*). The syringe was inserted through the membrane of the caps and sampled the volatiles for 30 min at 45°C; subsequently, metabolites were desorbed for 30 s at 240C and baked for an additional 4.5 min in the injection port. The gas chromatograph was fitted with a microchannel plate (MCP) detector. The temperature program of the column was as follows: 40°C for 5 min, 2 °C a min for 17.5 min followed by 25 °C a min for 10 min. A final hold time of

5 min at 325°C was used. The carrier gas was helium. A split ratio of 250 was used based on better peak resolution. An internal standard of cineole (Sigma-Aldrich, St. Louis, MO, USA) was run with each sample and used to compute relative abundances. The mass detector was in the range of 40 to 650 m/z.

To analyze the data, MassLynx software was used. The response threshold was set to an absolute area of 10.00. The software automatically picked out peaks and computed peak areas. To obtain a relative quantification, peaks were compared across samples and normalized to the internal standard. Peaks were first searched against the NIST5 database to identify potential hits. Most potential metabolites were confirmed by a standard mixture in 50% AJM. The standard mixtures are in *Supplementary file 4*.

## Chemical complementation of co-culture containing A. pomorum adhA

A co-culture containing *S. cerevisiae* and *A. pomorum* WT or *A. pomorum adhA* was grown for 72 hr before use in the T-maze. For the physiological concentrations of acetate-derived metabolites, concentrations were added as in *Table 2* and then mixed 1:1 with water prior to behavioral analysis. For the acetaldehyde metabolic derivatives chemical complementation group, a 1:1 mixture of the mutant co-culture: water was supplemented with 1,1-diethoxyethane, 2,3-butanedione, and acetoin at final concentrations of 0.01%, 0.15% and 0.15%, respectively and used immediately in the T-maze. Acetic acid and/or acetaldehyde were added to the culture, allowed to sit at RT for 35 min, mixed 1:1 with water and then placed into the T-maze vials.

Standard curves were used to calculate the concentrations of individual metabolites (*Table 2—source data 2*, *3*, *5* and *6*). The standard curves were generated on two separate experiments in which 3 concentrations of each standard was used. The concentrations of the metabolites were independently calculated from the standard curve equations generated on the two separate days. Estimated concentrations from each standard curve equation were averaged (*Table 2*). The experimental data are based on the peak areas of the *S. cerevisiae-A. malorum* co-culture.

## Ester, acid, and acetaldehyde metabolic derivative mixture

The 9-metabolite mixture contains 1.5% acetic acid, 0.0003% isoamyl acetate, 0.0003% 2-phenethyl acetate, 0.01% ethyl acetate, 0.002% ethyl lactate, 0.3% 1,1-diethoxyethane, 0.3% 2,3-butanediol, 0.3% 2,3-butanedione, and 0.3% acetoin in filtered milliQ water.

## *Drosophila* survival in the presence of ethanol and acetic acid

Adult male flies (0–3 d-old) were collected and matured for one day on fly food. Flies were then placed into vials containing kimwipes with 5 mL of either Shields and Sang Insect Medium (Sigma, St. Louis, MO; positive control), MilliQ water (negative control), or MilliQ water with ethanol (9.4%), acetic acid (3.42%), or ethanol and acetic acid (1.4% and 2.8% respectively). Survival was assessed every 12 hr for 7 d. For each condition 5 mL was given at 0 and 12 hr and every 24 hr thereafter. Experimental replicates were considered separate vials (5–6 per group). Each replicate contained 8–31 flies.

## Egg-laying preference assay

Egg-preference assay was adapted from Joseph *et al* 2009 (*Joseph et al., 2009*). Microbial cultures grown for 96 hr were heated to 65°C for 10 min, mixed 1:1 with 1.6% agarose and poured into a 35×10 mm polystyrene tissue culture dish (Fisher Scientific, PA, USA) separated in two by a straight-edge razor blade. Flies were starved for ~18 hr prior to the experiment. The 35 mm petri dish was placed within clear flat top boxes with dimensions 2 5/16" X 2 5/16" X 5 1/16" (TAP plastics, San Leandro, CA, USA). The test and control sides were alternated for each replicate. *Drosophila* aged 4–10 days (n = 50–100) was allowed to lay eggs for 8 hr. After the assay, the number of eggs on deposited on each choice was tabulated and an egg-laying index was computed analogously to the olfactory response index.

## *Drosophila* development in co-cultures containing S. cerevisiae and either A. pomorum wild-type or adhA co-cultures with environmental exposure

0–3 d-old *Drosophila* were collected (40 females and 15 males per tube). After three days, each tube of flies was placed into six oz. polypropylene square bottom *Drosophila* bottles (Dot Scientific Inc., MI, USA) in which a 35×10 mm polystyrene tissue culture dish (Fisher Scientific, PA, USA) was fitted inside the opening hole. The culture dish contained either the co-culture with *A. pomorum* wild-type and *S. cerevisiae* or the co-culture with *A. pomorum adhA* and *S. cerevisiae*. *Drosophila* was allowed to lay eggs for 4 hr. The co-cultures were grown for 72 hr at 28 C, 200 rpm. The cultures were mixed 1:1 with 1.6% agarose and 4 mL was poured into each 35 mm culture dish. The eggs were counted manually immediately following the 4 hr time window of egg-laying. The plates were placed into an incubator at 60% humidity and 25C on a 12:12 hr light dark cycle. After three days, the plates were exposed to the environment by placing them on the floor with their lids off for 10 min. Subsequently, total pupae and adults were counted daily.

In the case of no open environmental exposure (*Figure 7B*), the co-culture was mixed with 1.6% agarose 1:1. 8 mL was distributed into narrow polypropylene fly vials (28.5×95 mm, VWR, PA, USA). After 4 hr, adults were removed and the eggs were placed in an incubator at 60% humidity and 25C on a 12:12 hr light dark cycle.

### Data analysis

Data analysis was performed in Prismv6.0b. Specific statistical tests are noted for individual experiments. In behavioral experiments, a Shapiro-Wilk normality test determined whether the underlying data were consistent or inconsistent with a normal distribution. If consistent, a parametric test was used to evaluate differences; if inconsistent, a non-parametric test was used.

## Acknowledgements

We thank Dr. Fabian Staubach (Stanford University, USA), Dr. Angus Chandler (University of California, Berkeley, USA), Dr. Matthew Goddard (The University of Auckland, New Zealand), Dr. Dan Jarosz (Stanford University, USA) and the Phaff Yeast Culture Collection (UC Davis, USA) for microbial strains. We thank Dr. Ryan Joseph and Dr. Karen Menuz (Yale University/University of Connecticut) for help with the *Drosophila* behavioral experiments; the Yale West Campus Analytical Core for use of the GC-MS; Dr. Terence Wu and Dr. Eric Patridge (Yale University, USA) for advice and experimental help with the GC-MS; Dr. Scott Strobel and Michelle Legaspi for use of and help with SPME GC-MS; Dr. John Carlson (Yale University, USA) for the T-maze and several *Drosophila* lines. We also thank Dr. Won Jae Lee (Seoul National University, S. Korea) for the *A. pomorum* WT and *adhA* strains, and John Carlson, Craig Crews, and Andrew Goodman (Yale University, USA) for helpful discussions. This work was supported by National Institutes of Health Grants NIDDKRC12T32GM007499-36, 5T32HG003198-10, 1 R01GM099563, 7RC1DK086831. The raw data for the metabolomics experiments can be found at https://figshare.com/account/home#/projects/4735.

## Additional information

### Funding

| Funder | Grant reference number | Author |
|---|---|---|
| NIH Office of the Director | 5T32HG003198-10 | Caleb Fischer<br>Jo Handelsman |
| NIH Office of the Director | 1R01GM099563 | Jo Handelsman |
| NIH Office of the Director | NIDDKRC12T32GM007499-36 | Caleb Fischer |
| NIH Office of the Director | 7RC1DK086831 | Jo Handelsman |

The funders had no role in study design, data collection and interpretation, or the decision to submit the work for publication.

### Author contributions
CNF, Conception and design, Acquisition of data, Analysis and interpretation of data, Drafting or revising the article; EPT, Acquisition of data, Analysis and interpretation of data, Drafting or revising the article; JMC, EVS, JH, NAB, Conception and design, Analysis and interpretation of data, Drafting or revising the article

### Author ORCIDs
Caleb N Fischer, http://orcid.org/0000-0003-4223-8511
Jason M Crawford, http://orcid.org/0000-0002-7583-1242
Nichole A Broderick, http://orcid.org/0000-0002-6830-9456

## Additional files

### Supplementary files
• Supplementary file 1. Drosophila melanogaster stocks used in experiments.

• Supplementary file 2. Microorganisms used in experiments and their sources

• Supplementary file 3. Chemicals or solutions used in T-maze and GC-MS experiments.

• Supplementary file 4. Metabolite mixture concentrations used for identification and quantification in SPME GC-MS.

• Supplementary file 5. Composition of metabolite mixtures 1-21 used in *Figure 6*.

### Major datasets
The following dataset was generated:

| Author(s) | Year | Dataset title | Dataset URL | Database, license, and accessibility information |
|---|---|---|---|---|
| Fischer, CN, E Trautman, JM Crawford, EV Stabb, NA Broderick, J Handelsman | 2016 | Data from: Metabolite exchange within the microbiome produces compounds that influence Drosophila behavior | https://figshare.com/projects/Metabolite_exchange_within_the_microbiome_produces_compounds_that_influence_Drosophila_behavior/4735 | Publicly available at Figshare (https://figshare.com) |

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
