## [Decision Letter]

Thank you for submitting your article "Metabolite exchange within the microbiome produces compounds that influence *Drosophila* behavior" for consideration by *eLife*. Your article has been favorably evaluated by Wendy Garrett (Senior Editor) and three reviewers, one of whom is a member of our Board of Reviewing Editors. The following individual involved in review of your submission has agreed to reveal his identity: Michael B Eisen (Reviewer #2).

The reviewers have discussed the reviews with one another and the Reviewing Editor has drafted this decision to help you prepare a revised submission.

Summary:

The manuscript examines olfactory attraction to microbes found in the *Drosophila* microbiome. The authors show that a co-culture of yeast and bacteria (*S. cerevisiae* and *A. malorum*) grown together is more attractive than when the two cultures are mixed before testing. Attraction is reduced in Or42b and OrCo mutants with olfactory defects. Ethanol reduction and acetic acid production are unique to the co-culture. The preference toward the co-culture can largely be explained by acetic acid and esters production. The general point – that microbes interact with each other to affect the culture they are growing in, and that this in turn affects the way the culture is perceived by other organisms – is almost certainly true, and is interesting and important. We think that this study is of interest to the broad readership of *eLife* and support the publication of a streamlined and improved version which addresses the major points raised.

Essential revisions:

1) The authors change the statistical analyses throughout the manuscript without clarifying why they do so. It is not clear why the authors sometimes choose to compare if the value differs from 0 versus an ANOVA with a post-hoc test. In general, when making statements comparing conditions (to say that effect increases etc.) this cannot be based on a comparison vs. 0 and based on the magnitude of the p-value. In general, the statistical tests should be non-parametric when dealing with behavioural indices.

2) The interchange between microbiome and microbes is confusing. The authors are testing olfactory preference to microbes found in the microbiome but they are not studying how the microbiome affects behavior. It is not clear that *S. cerevisiae* and bacteria reside in the same niche or interact in the host. Other studies have shown that there is no matching between yeast and microbes associated with wild *Drosophila*. The title and text should be changed to use the word microbe rather than microbiome. The authors should be careful to distinguish what this work shows about microbial interactions versus the microbiome.

3) The descriptions in the Methods are incomplete. It is not clear how the cultures are presented to the flies, whether there was air flow, the possibility that the flies could touch the substrate etc.

4) Why was the comparison of co-cultures to mixtures of individual species made after they had reached stationary phase? *Drosophila* prefer exponentially growing cultures to stationary phase cultures, and their preference varies considerably as a function of cell density and growth rate. These are very difficult experiments to get right, but why didn't the authors didn't compare the growing co-cultures to growing mono-cultures?

5) The GC-MS data is surprising, especially the report that isoamyl acetate is unique to the co-culture. Previous studies found that this compound is produced in large quantities by many different strains (wild and lab) of *S. cerevisiae*. It's obviously possible that this result is a product of the different strains and culture conditions used in this paper. But the failure of yeast alone to produce this here warrants some discussion.

6) The preference of the flies for the co-culture seems to depend on the presence of specific olfactory receptors and it is proposed that OrCo and Or42b are key for mediating this attraction. Unfortunately, these findings require validation by more stringent genetic experiments. Currently, the relevant genetic background controls are missing (e.g. heterozygotes). Also, the involvement of Or42b and OrCo should be validated using different approaches (for example, by neuronal silencing of the neurons expressing these receptors or rescue experiments).

7) What is the logic of the survival experiments? Why look at adults? If there really is a selective advantage for flies in calibrating the specific balance of ethanol and acetic acid, then surely the relevant life stage to look at is eggs and larvae, not adults. Adults can come in and lay eggs and then leave (as they often do) – eggs and larvae don't have that option. And as the authors point out here, cultures like this are not stable. If selection has indeed driven flies to choose ideal ethanol and acetic acid concentrations, then the flies should be selecting egg laying conditions that will be good for eggs and larvae over the coming days, not those that are good for adults at that exact concentration.

8) The authors go to a great length to describe the interesting findings buried in the supplemental figures to Figure 6. We suggest that they restructure the manuscript to include them as main figures in order to ensure that the reader can properly follow the text. The metabolic reconstitution experiments are interesting and highly relevant. Currently they are difficult to grasp as they follow a lot of data and the data are not easily accessible.

---

## [Author Response]

*Essential revisions:*

*1) The authors change the statistical analyses throughout the manuscript without clarifying why they do so. It is not clear why the authors sometimes choose to compare if the value differs from 0 versus an ANOVA with a post-hoc test. In general, when making statements comparing conditions (to say that effect increases etc.) this cannot be based on a comparison vs. 0 and based on the magnitude of the p-value. In general, the statistical tests should be non-parametric when dealing with behavioural indices.*

We appreciate the feedback from the reviewers and we have revised the statistics as suggested and described below:

We first use a Shapiro-Wilk analysis to see if the underlying data is consistent or inconsistent with a normal distribution. If consistent, we perform a parametric test. If inconsistent, we perform a non-parametric test.

When asking whether *Drosophila* is attracted, neutral, or repelled by a condition in an absolute sense (not relative to any other group) we use either the parametric Wilcoxon-signed-rank test in which the median for each group is compared to a hypothetical value of zero or the parametric one sample t-test in which the group mean is compared against a hypothetical value of zero.

When asking whether *Drosophila* prefers one condition relative to another, we use the non-parametric Mann Whitney test when 2 groups are compared or the Kurskal Wallis when >2 groups are compared (with post-hoc comparisons). The parametric equivalents were used when appropriate—the unpaired two-sample t-test for 2 groups and the one way ANOVA for >2 groups (with post-hoc comparison).

*2) The interchange between microbiome and microbes is confusing. The authors are testing olfactory preference to microbes found in the microbiome but they are not studying how the microbiome affects behavior. It is not clear that S. cerevisiae and bacteria reside in the same niche or interact in the host. Other studies have shown that there is no matching between yeast and microbes associated with wild Drosophila. The title and text should be changed to use the word microbe rather than microbiome. The authors should be careful to distinguish what this work shows about microbial interactions versus the microbiome.*

The microorganisms used in the study were isolated from *Drosophila* species (*S. cerevisiae* was isolated from *Drosophila pinicola* and *A. malorum* was isolated from *D. melanogaster*, [[Supplementary-material SD34-data]]). *Saccharomyces* and *Acetobacter* also inhabit fruit fermentations (1), but organisms that cycle between host and environment are still microbiome members and previous work has shown that *Drosophila* regularly replenish their microbiome from food substrates. We believe that referring to the microorganisms used in this study as microbes neglects their known ecological niches. For these reasons, we argue that the use of microbiome is warranted. However, we have modified the title and sentences within the text to more clearly indicate that our study is exploring fly attraction to interactions between microbiome members.

*3) The descriptions in the Methods are incomplete. It is not clear how the cultures are presented to the flies, whether there was air flow, the possibility that the flies could touch the substrate etc.*

We have clarified the T-maze assay in the Methods to address the concerns specified:

· We specified how the cultures are presented to the flies.

· We clarified that no air flow was used.

· We stated that in observations during troubleshooting experiments (using red light) the flies rarely reached the substrate. Instead the flies moved away from the center and stopped before reaching the substrate.

*4) Why was the comparison of co-cultures to mixtures of individual species made after they had reached stationary phase? Drosophila prefer exponentially growing cultures to stationary phase cultures, and their preference varies considerably as a function of cell density and growth rate. These are very difficult experiments to get right, but why didn't the authors didn't compare the growing co-cultures to growing mono-cultures?*

The times were chosen because we reasoned that the cultures are highly dynamic during exponential growth, making it very difficult to ensure that the species/strains are in the exact same phase. We did test whether our finding that *Drosophila* prefers the co-culture to the separate culture was specific for a single timepoint or persisted over time (Figure 2). Specifically, we compared the co-cultures to the mixture of individual species beginning at (34 h) and periodically all the way to 127 h after inoculation. *A. malorum* reaches stationary phase at ~34 h, the first timepoint tested, and *S. cerevisiae* reaches stationary phase at ~24h, indicating that in the early time tested, the strains in the separate-culture mixture were either in early stationary phase or late exponential phase.

However, as the reviewers note, we did not compare the earlier timepoints of the separate cultures (or separate-culture mixture) to the co-culture during ethanol catabolism. A previous study found that *Drosophila* was more attracted to a 1-d vs. 3-d culture of *S. cerevisiae* (3), although we are unaware of the exact relationship between 1- and 3-d cultures and exponential and stationary phase. This result may suggest, as reviewers note, that *Drosophila* may not prefer theco-culture if it was pitted against earlier aged (e.g. 1 d) mono-cultures or separate-culture mixtures. We performed an experiment to address the reviewers’ concern. The experiment has been incorporated into Figure 2—figure supplement 1 and Eand is described in the second paragraph of the Results.

*5) The GC-MS data is surprising, especially the report that isoamyl acetate is unique to the co-culture. Previous studies found that this compound is produced in large quantities by many different strains (wild and lab) of S. cerevisiae. It's obviously possible that this result is a product of the different strains and culture conditions used in this paper. But the failure of yeast alone to produce this here warrants some discussion.*

We reason that low production of esters with our yeast strain is most likely explained by low nitrogen content in our media. Our medium was apple juice based, which may be lower in nitrogen compared with many standard laboratory media. Nitrogen availability is positively correlated with ester production in *S. cerevisiae* (2, 3).

We have addressed by including a paragraph (third) in the Discussion in which we:

· Discuss how fruit nitrogen availability could determine which microorganisms produce attractive esters.

- Fruits with high nitrogen content may lead to ester production by yeasts.

- Low nitrogen content may indicate that ester emission by yeasts.

· In low-nitrogen fruit environments, *Acetobacter* can produce esters (as evidenced by our results).

- This could be due to a different mechanism of ester production than yeast (e.g. in low nitrogen) or

- Its ability to assimilate nitrogen from acetic acid-mediated killing of yeasts.

*6) The preference of the flies for the co-culture seems to depend on the presence of specific olfactory receptors and it is proposed that OrCo and Or42b are key for mediating this attraction. Unfortunately, these findings require validation by more stringent genetic experiments. Currently, the relevant genetic background controls are missing (e.g. heterozygotes). Also, the involvement of Or42b and OrCo should be validated using different approaches (for example, by neuronal silencing of the neurons expressing these receptors or rescue experiments).*

Our analysis tested homozygous mutants in comparison to wild-type control (the mutants were outcrossed to the wild-type controls). We performed an additional experiment (new Figure 3) in which the heterozygotes of each mutant and an additional homozygous mutant of OrCo were tested.

*7) What is the logic of the survival experiments? Why look at adults? If there really is a selective advantage for flies in calibrating the specific balance of ethanol and acetic acid, then surely the relevant life stage to look at is eggs and larvae, not adults. Adults can come in and lay eggs and then leave (as they often do) – eggs and larvae don't have that option. And as the authors point out here, cultures like this are not stable. If selection has indeed driven flies to choose ideal ethanol and acetic acid concentrations, then the flies should be selecting egg laying conditions that will be good for eggs and larvae over the coming days, not those that are good for adults at that exact concentration.*

We chose to look at survival experiments because we thought survival was a proxy for fitness associated with consuming the major metabolites associated with the co-culture (ethanol and acetic acid). We chose to examine adults because adults were the ones that preferred the co-culture and one underlying reason they might prefer the co-culture is due to the concentration of specific metabolites. Adults are doing more than laying eggs. They are consuming the substrate, so it seems reasonable to us to test whether they were attracted to a substrate that increased their fitness. Furthermore, the timepoint at which *Drosophila* was most attracted to the co-culturecorrelated with low ethanol concentrations, which are known to benefit flies.

To address the reviewers’ critique, we expanded our experiment that showed that *Drosophila* preferred to lay eggs in the co-culture with *A. pomorum* WTversus the co-culture with *A. pomorum adhA* by looking at the consequences of larvae developing in the co-culture with either *A. pomorum* WT or *A. pomorum adhA*.

We first used larval development time (time to pupation) as a proxy for nutrition. Next, we exposed the eggs that had been laid in the co-culture containing *A. pomorum* WT or *A. pomorum adhA* to environmental microorganisms. (see new Figure 7 in paper).

*8) The authors go to a great length to describe the interesting findings buried in the supplemental figures to Figure 6. We suggest that they restructure the manuscript to include them as main figures in order to ensure that the reader can properly follow the text. The metabolic reconstitution experiments are interesting and highly relevant. Currently they are difficult to grasp as they follow a lot of data and the data are not easily accessible.*

We have taken the reviewers’ suggestions and edited the manuscript to include some of the reconstruction experiments into Figure 6 (in the main text).

References:

1) Barata A, Malfeito-Ferreira M, Loureiro V. 2012. The microbial ecology of wine grape berries. Int J Food Microbiol. 153(3):243–59.

2) Rollero S, Bloem A, Camarasa C, Sanchez I, Ortiz-Julien A, et al. 2015. Combined effects of nutrients and temperature on the production of fermentative aromas by Saccharomyces cerevisiae during wine fermentation. Appl Microbiol Biotechnol. 99(5):2291–2304.

3) Schiabor KM, Quan AS, Eisen MB. 2014. Saccharomyces cerevisiae mitochondria are required for optimal attractiveness to Drosophila melanogaster. PLoS ONE. 9(12):e113899.